# Focusing on the Riskiest: Gaussian Mixture Models for Safe Reinforcement Learning

## Abstract

Reinforcement learning under safety constraints remains a fundamental challenge. While primal–dual formulations provide a principled framework for enforcing such constraints, their effectiveness depends critically on accurate modeling of cost distributions. Existing approaches often impose Gaussian assumptions and approximate risk either by the mean or by CVaR, yet these formulations inherently fail to capture complex, multimodal, or heavy-tailed risks. To overcome these limitations, we propose **GMM-SSAC** (Gaussian Mixture Model-Based Supremum CVaR-Guided Safe Soft Actor-Critic), whose core is the Supremum Conditional Value-at-Risk (SCVaR) criterion: a coherent and robust safety measure that explicitly targets the worst-case tail across all components of a Gaussian mixture. To support accurate SCVaR estimation online, we introduce an incremental EM-based update that refines the GMM parameters by blending instantaneous safety samples with Bellman-transformed estimates—ensuring unbiased, convergent parameter estimates for reliable SCVaR computation. Empirical evaluations on standard safety benchmarks demonstrate that GMM-SSAC substantially improves risk sensitivity and safety while maintaining competitive task performance, validating SCVaR as a principled and effective cost estimator for safe reinforcement learning.

## 1 Introduction

Safe Reinforcement Learning (Safe RL) aims to enable autonomous agents to learn effective policies while satisfying safety constraints. With RL increasingly applied in safety-critical domains such as healthcare (Yu et al., 2021), robotics (Tang et al., 2024), finance (Hambly et al., 2023), and autonomous driving (Kiran et al., 2021), ensuring safe and reliable operation has become crucial. Standard RL methods, which focus on maximizing cumulative rewards, often overlook risks and safety violations during training, leading to unsafe behaviors and potentially catastrophic failures. This highlights the need for principled frameworks that explicitly integrate safety into the learning process.

A principled framework for Safe RL is the Constrained Markov Decision Process (CMDP) (Altman, 2021), where the agent maximizes rewards subject to cost constraints. Primal–dual

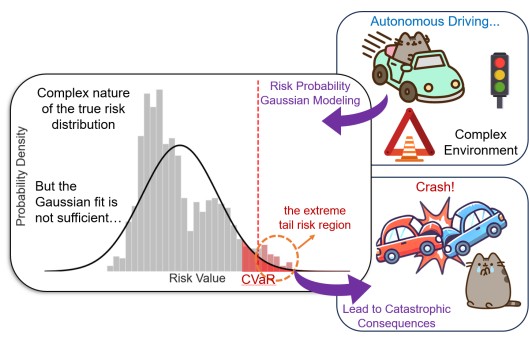

Figure 1: Demonstration of how Gaussian-based CVaR's underestimation of tail risks can lead to dangerous driving behaviors, as the model fails to properly account for extreme events that deviate from normal distribution assumptions.

methods are commonly used to solve CMDPs (Achiam et al., 2017; Tessler et al., 2018; Chow et al., 2018), with a safety critic estimating the distribution of cumulative costs, where "cost" denotes penalties from safety violations or undesirable actions. To better capture risk, prior studies have introduced measures such as Conditional Value-at-Risk (CVaR) (Tamar et al., 2015; Chow et al.,

2018; Coache et al., 2023), which focuses on tail losses, and upper confidence bounds (UCBs) (Wu et al., 2024), which provide conservative cost estimates.

However, most existing approaches approximate long-term costs with a single Gaussian, which is often too simplistic for safety-critical settings. As shown in Fig. 1, the empirical cost distribution can be complex and multi-modal, while its Gaussian fit fails to capture tail behavior, leading CVaR to underestimate extreme risks and induce hazardous policies. This motivates the need for more expressive distributional models. Gaussian Mixture Models (GMMs) provide a natural choice: they have universal approximation capability for continuous distributions (Chacko & Viceira, 2003; Jalali et al., 2019) and can represent distinct safety-critical modes through different components, offering both flexibility and interpretability.

GMMs have also been applied in RL for Q-function approximation (Agostini & Celaya, 2010; Vu & Slavakis, 2024), improving function accuracy, sample efficiency, and robustness in non-stationary environments. However, these works do not address the critical issue of risk modeling in Safe RL. To fill this gap, we introduce the **Supremum Conditional Value-at-Risk (SCVaR)**, defined as the maximum CVaR across GMM components. SCVaR is a coherent risk measure (Artzner et al., 1999), providing a conservative estimate that explicitly accounts for the worst-case risks captured by each component. To estimate GMM parameters robustly, we design an incremental EM refinement that blends Bellman-updated samples with new observations (Moon, 1996), ensuring unbiased online updates. Integrating this safety critic into the Soft Actor-Critic (SAC) framework (Haarnoja et al., 2018), we obtain the proposed **GMM-SSAC** algorithm.

Our contributions are threefold: (1) introducing SCVaR as a coherent risk measure that captures worst-case tail risks across mixture components; (2) developing an incremental EM-based update for accurate online GMM estimation; and (3) designing GMM-SSAC, which achieves improved safety with competitive performance on standard benchmarks.

## 2 PRELIMINARIES

### 2.1 CONSTRAINED MARKOV DECISION PROCESSES

A CMDP extends the standard MDP by incorporating safety constraints into the optimization framework. Formally, a CMDP is defined as a tuple $\mathcal{M} = (\mathcal{S}, \mathcal{A}, P, r, c, \gamma, D)$, where $\mathcal{S}$ and $\mathcal{A}$ denote the state and action spaces, $P(s'|s, a)$ is the transition probability from state $s$ to state $s'$ given action $a$, $r(s, a)$ is the reward function, and $c(s, a)$ is the cost function associated with safety risks. The discount factor $\gamma \in [0, 1)$ balances immediate and future returns. The scalar $D$ represents the threshold on the expected cumulative cost, defining the safety constraint.

The goal of a CMDP is to find a policy $\pi$ that maximizes the expected cumulative reward while ensuring that the cumulative cost remains below the threshold $D$. This constrained optimization problem is formulated as:

$$\max_{\pi} \quad \mathbb{E}_{(s_t, a_t) \sim \rho_\pi} \left[ \sum_{t=0}^{\infty} \gamma^t r(s_t, a_t) \right], \quad \text{s.t.} \quad \mathbb{E}_{(s_t, a_t) \sim \rho_\pi} \left[ \sum_{t=0}^{\infty} \gamma^t c(s_t, a_t) \right] \leq D, \qquad (1)$$

where $\rho_\pi$ is the state-action distribution induced by the policy $\pi$. The inequality ensures that the policy satisfies the specified safety requirement.

### 2.2 PRIMAL–DUAL METHOD FOR CMDPS

To solve the constrained optimization in Eq. 1, a standard approach is to adopt the primal–dual method, which introduces a non-negative Lagrange multiplier $\lambda$ associated with the safety constraint. The resulting Lagrangian is given by:

$$\mathcal{L}(\pi, \lambda) = \mathbb{E}_{(s_t, a_t) \sim \rho_\pi} \left[ \sum_{t=0}^{\infty} \gamma^t r(s_t, a_t) \right] - \lambda \left( \mathbb{E}_{(s_t, a_t) \sim \rho_\pi} \left[ \sum_{t=0}^{\infty} \gamma^t c(s_t, a_t) \right] - D \right). \qquad (2)$$

The optimization then proceeds by solving the following saddle-point problem:

$$\max_{\pi} \min_{\lambda \geq 0} \mathcal{L}(\pi, \lambda). \qquad (3)$$

Intuitively, the policy $\pi$ (primal variable) is updated to maximize the Lagrangian objective, while the multiplier $\lambda$ (dual variable) is adjusted to penalize constraint violations. In practice, this leads to an iterative update scheme where policy optimization and dual variable adjustment are alternated, ensuring that the learned policy balances task performance and safety satisfaction.

## 2.3 Conditional Value at Risk

Conditional Value at Risk (CVaR) is a coherent risk measure that captures the expected loss in the tail of a distribution. For a random variable $X$, the Value at Risk (VaR) at confidence level $\alpha \in (0, 1]$ is defined as

$$\text{VaR}_\alpha(X) = \inf\{x \in \mathbb{R} \mid F_X(x) \geq 1 - \alpha\}, \tag{4}$$

where $F_X$ denotes the cumulative distribution function of $X$.

The CVaR at level $\alpha$ is formally defined, for distributions that are absolutely continuous, as

$$\text{CVaR}_\alpha(X) = \mathbb{E}[X \mid X \geq \text{VaR}_\alpha(X)]. \tag{5}$$

For general distributions, equivalent formulations can be obtained through integral representations of the quantile function (see, e.g., Rockafellar & Uryasev (2000)), but in this work we focus on the Gaussian case, where absolute continuity holds.

For Gaussian random variables $X \sim \mathcal{N}(\mu, \sigma^2)$, the CVaR has the closed-form expression

$$\text{CVaR}_\alpha(X) = \mu + \sigma \cdot \frac{\phi(\Phi^{-1}(1 - \alpha))}{1 - \alpha}, \tag{6}$$

where $\phi(\cdot)$ and $\Phi(\cdot)$ are the PDF and CDF of the standard normal distribution, respectively. Here, $\alpha$ controls the degree of risk aversion: smaller values emphasize extreme tail risks, whereas larger values consider broader outcomes.

Several works (Yang et al., 2021; Wu et al., 2024) have extended the primal–dual formulation of CMDPs by replacing the expectation of cumulative costs in the Lagrangian (Eq. 2) with a CVaR-based risk term. Specifically, the modified Lagrangian becomes

$$\mathcal{L}_{\text{CVaR}}(\pi, \lambda) = \mathbb{E}_{(s_t, a_t) \sim \rho_\pi}\left[\sum_{t=0}^{\infty} \gamma^t r(s_t, a_t)\right] - \lambda\left(\text{CVaR}_\alpha\left(\sum_{t=0}^{\infty} \gamma^t c(s_t, a_t)\right) - D\right). \tag{7}$$

This CVaR-based primal–dual formulation provides a more conservative alternative to expectation-based constraints by explicitly regularizing the tail of the cost distribution.

## 3 Methodology

### 3.1 Supremum Conditional Value at Risk

Fig. 1 illustrates the limitations of single Gaussian approximations in capturing complex safety cost distributions. To overcome this challenge, we propose a more expressive distributional framework using GMMs, defined as:

$$\mathcal{G}^\pi(s, a) \approx \sum_{k=1}^{K} \omega_k \mathcal{N}(\mu_k, \sigma_k^2), \tag{8}$$

where $\mathcal{G}^\pi(s, a)$, represents the probabilistic distribution of cumulative safety costs $\sum_{t=0}^{\infty} \gamma^t c(s_t, a_t)$ under the policy $\pi$, $\omega_k$ are the mixing coefficients satisfying $\sum_{k=1}^{K} \omega_k = 1$, and $\mathcal{N}(\mu_k, \sigma_k^2)$ represents the $k$-th Gaussian component with mean $\mu_k$ and variance $\sigma_k^2$.

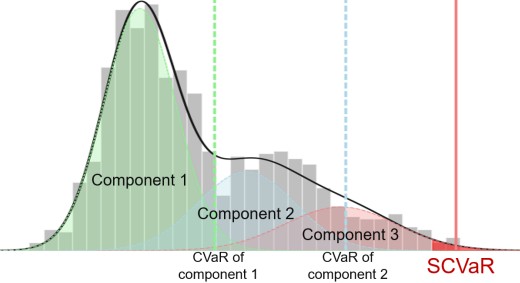

Figure 2: SCVaR illustration showing how it identifies the worst-case risk (red line) beyond individual CVaR components (green and blue) in a GMM.

To quantify extreme tail risks in GMMs, we introduce the concept of SCVaR as shown in Fig. 2. SCVaR extends the traditional CVaR framework by focusing on the worst-case tail risk among all components in a multimodal distribution. We define SCVaR as follows.

**Definition 1** (Supremum Conditional Value at Risk). *SCVaR is the maximum CVaR across all components of the GMM, capturing the worst-case tail risk:*

$$SCVaR_\alpha = \sup_{k \in \{1,\dots,K\}} CVaR_\alpha^{(k)}, \tag{9}$$

*where $CVaR_\alpha^{(k)}$ is the CVaR at level $\alpha$ of the $k$-th Gaussian component, and $K$ is the total number of components.*

**Theorem 1** (Proof in Appendix A.1). *SCVaR provides a conservative upper bound on the mixture CVaR of a GMM, i.e.,*

$$\text{SCVaR}_\alpha \geq \text{CVaR}_\alpha^{\text{GMM}},$$

*where $\text{CVaR}_\alpha^{\text{GMM}}$ denotes the CVaR of the full mixture at level $\alpha$.*

Importantly, the theorem highlights that mixture CVaR is always a *convex combination* of component-wise CVaRs, and therefore can underestimate rare but high-risk modes. SCVaR avoids this structural limitation by explicitly preserving the worst-component tail, making it the tightest coherent upper bound suitable for multimodal safety constraints.

**Proposition 1** (Proof in Appendix A.2). *SCVaR is a coherent risk measure; it satisfies monotonicity, translation invariance, positive homogeneity, and subadditivity, ensuring compatibility with primal–dual CMDP optimization.*

To further clarify the behavioral difference between SCVaR and CVaR in multimodal settings, we include a simple investment portfolio example in Appendix A.3, where mixture CVaR understates catastrophic components while SCVaR correctly identifies the worst-case tail.

### 3.2 INCREMENTAL UPDATING SAFETY CRITIC WITH BELLMAN OPERATOR

Our next challenge is to estimate the parameters $\{(\mu_k, \sigma_k, \omega_k)\}_{k=1}^K$ in $\mathcal{G}^\pi(s, a)$. Unlike traditional distribution parameter estimation problems, in the CMDP environment, since the safety cost is the discounted sum of the instantaneous safety measures over time, we cannot directly obtain samples of the cost. The only data we can collect are the instantaneous safety measures. To overcome these challenges, we employ an incremental updating approach that blends instantaneous safety measures from the Bellman operator with historical distribution estimates.

**1) Sampling Operation $\mathcal{X}$.**

We first sample from the existing parameter estimates, which results in a sampling operation:

$$\mathcal{X}(\mathcal{G}^\pi(s, a), N) = \bigcup_{k=1}^K \mathcal{X}_k(\mathcal{N}(\mu_k, \sigma_k^2), N_k), \tag{10}$$

where $\mathcal{X}_k$ denotes $N_k$ samples drawn independently from the $k$-th Gaussian component. The allocation of $N_k$ is calculated as:

$$N_k = \lfloor \omega_k N \rfloor, \quad \forall k \in \{1, \dots, K\}. \tag{11}$$

If $\sum_{k=1}^K N_k \neq N$ due to rounding, the remaining samples are assigned to components with the highest $\omega_k$. We denote the final sample set as $\Psi(s, a) = \mathcal{X}(\mathcal{G}^\pi(s, a), N)$.

**2) Bellman Sampling with Operator $\mathcal{B}$.** Next, we use the Bellman Equation to generate a new estimate based on the instantaneous safety measures. For a given state-action pair $(s, a)$, the Bellman operator $\mathcal{B}$ updates the safety critic by combining the real-time observed safety cost $c(s, a)$ and the expected discounted future safety costs. Samples $x_i'$ are drawn from the target safety critic distribution $\mathcal{G}_{\text{target}}^\pi(s', a')$ using the sampling operator $\mathcal{X}$. The transformed sample set is then defined as:

$$\Psi_{\mathcal{B}}(s, a) = \{c(s, a) + \gamma x_i' \mid x_i' \in \mathcal{X}(\mathcal{G}_{\text{target}}^\pi(s', a'), M)\}, \tag{12}$$

where $M$ represents the total number of samples.

**3) Incremental Refinement with Operator $\mathcal{R}$.** The incremental refinement operator $\mathcal{R}$ blends the historical sample set $\Psi(s, a)$ with the Bellman-transformed sample set $\Psi_{\mathcal{B}}(s, a)$ using a weight

parameter $\beta$. Specifically, it updates the target sample set by sampling from each set according to the weights $1 - \beta$ and $\beta$, producing the refined set $\Psi_{\text{update}}(s, a)$. This blended update introduces an important stabilizing effect. If the update were to rely only on the most recent Bellman-transformed samples, the safety critic would be forced to track a rapidly changing target distribution, which often results in high-variance EM updates and unstable GMM parameters. Incorporating historical samples provides a smoothing effect similar to Polyak averaging or target networks, which helps maintain temporal continuity of the learned distribution. The parameter $\beta$ controls the balance between stability and adaptability, with larger values giving more emphasis to new Bellman-based estimates and smaller values preserving more of the previously learned distribution.

**4) EM Estimation (Projection $\mathcal{P}$).** The EM algorithm (Moon, 1996) is used to estimate the GMM parameters from samples in $\mathcal{R}$, alternating between the following steps:

**E-step:** Compute the responsibility of each Gaussian component $k$ for each sample $x_m \in \Psi_{\text{update}}(s, a)$. Specifically:

$$\gamma_{mk} = \frac{\omega_k \varphi(x_m \mid \mu_k, (\sigma_k)^2)}{\sum_{j=1}^K \omega_j \varphi(x_m \mid \mu_j, (\sigma_j)^2)}, \tag{13}$$

where $\varphi(x_m \mid \mu_k, (\sigma_k)^2)$ is the Gaussian density function for the $k$-th component, defined as:

$$\varphi(x_m \mid \mu_k, (\sigma_k)^2) = \frac{1}{\sqrt{2\pi}\sigma_k} \exp\left(-\frac{(x_m - \mu_k)^2}{2\sigma_k^2}\right). \tag{14}$$

**M-step:** Update the GMM parameters based on the computed responsibilities:

$$\mu_k^{\text{update}} = \frac{\sum_{m=1}^M \gamma_{mk} x_m}{\sum_{m=1}^M \gamma_{mk}}, \quad (\sigma_k^{\text{update}})^2 = \frac{\sum_{m=1}^M \gamma_{mk}(x_m - \mu_k^{\text{update}})^2}{\sum_{m=1}^M \gamma_{mk}}, \quad \omega_k^{\text{update}} = \frac{\sum_{m=1}^M \gamma_{mk}}{M}. \tag{15}$$

Finally, normalize the mixing coefficients $\omega_k^{\text{update}}$ to ensure they sum to 1:

$$\omega_k^{\text{update}} \leftarrow \frac{\omega_k^{\text{update}}}{\sum_{j=1}^K \omega_j^{\text{update}}}, \quad \forall k \in \{1, \ldots, K\}. \tag{16}$$

**5) Neural Network Update.** The neural network $\mathcal{F}$ for safety critic is updated by minimizing the MSE loss between predicted GMM parameters $(\mu_k, \sigma_k, \omega_k)$ and update parameters $(\mu_k^{\text{update}}, \sigma_k^{\text{update}}, \omega_k^{\text{update}})$.

By executing these steps in sequence, we obtain the Safety Critic with Mixture Gaussian Representation (SC-MGR) algorithm (detailed in Alg. 2), and we establish its convergence.

### 3.2.1 CONVERGENCE ANALYSIS

We analyze the convergence of SC-MGR via the composite Bellman–EM operator $\mathcal{PT}^\pi$, where $\mathcal{T}^\pi$ is the distributional Bellman operator and $\mathcal{P}$ the EM-based projection onto the GMM space.

**Lemma 1.** *For any $\nu_1, \nu_2 \in \mathcal{P}_1(\mathbb{R})$, the set of probability measures on $\mathbb{R}$ with finite first moment, we have*

$$W_1(\mathcal{T}^\pi \nu_1, \mathcal{T}^\pi \nu_2) \leq \gamma W_1(\nu_1, \nu_2), \quad W_1(\mathcal{P}\nu_1, \mathcal{P}\nu_2) \leq W_1(\nu_1, \nu_2),$$

*so that*

$$W_1(\mathcal{PT}^\pi \nu_1, \mathcal{PT}^\pi \nu_2) \leq \gamma W_1(\nu_1, \nu_2), \quad \gamma \in (0, 1).$$

*Thus, $\mathcal{PT}^\pi$ is a $\gamma$-contraction on $\mathcal{P}_1(\mathbb{R})$.*

**Theorem 2.** *In a finite MDP with $|S|, |A| < \infty$ and $\gamma \in (0, 1)$, let the safety critic be modeled as a GMM and updated by $\mathcal{PT}^\pi$. Then:*

*1. (Bellman contraction) $\mathcal{T}^\pi$ is a $\gamma$-contraction in $W_1$ (Bellemare et al., 2017).*

*2. (EM projection non-expansiveness, proof in Appendix B) $\mathcal{P}$ is non-expansive in $W_1$, since EM matches first moments: $\mathbb{E}[\gamma_{mk}] = \omega_k \Rightarrow \mathbb{E}[\mu_k^{new}] = \mu_k$.*

*3. (Preservation of contraction) Combining (1) and (2), the composite operator $\mathcal{PT}^\pi$ is a $\gamma$-contraction.*

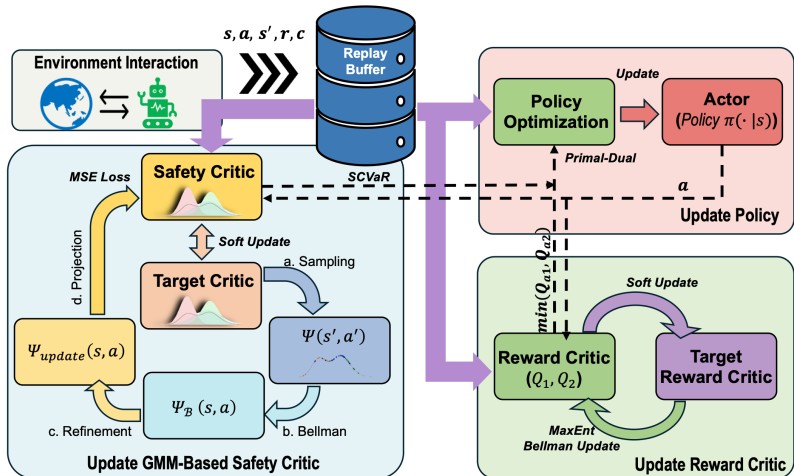

Figure 3: Framework overview of our proposed method. The architecture consists of three main components: (1) Update GMM-Based Safety Critic (left), which employs a cycle of sampling, Bellman updates, refinement, and EM projection operations to maintain the safety distribution; (2) Update Reward Critic (right-bottom), which follows the standard soft actor-critic architecture with twin Q-functions; and (3) Update Policy (right-top), which integrates both safety and reward signals through SCVaR-constrained policy optimization.

*Hence, SC-MGR converges to a unique fixed point $\mathcal{V}^* \in \mathcal{P}_1(\mathbb{R})$, defining the stable distributional safety critic.*

The key is that EM yields unbiased GMM parameter estimates, allowing the contraction mapping property of $\mathcal{T}^\pi$ (Tsitsiklis & Van Roy, 1996) to ensure convergence. In practice, the neural network only serves as a parametric approximation of the EM projection, and its MSE-based regression step does not change the Bellman–EM operator itself. Since the network performs small gradient updates toward the EM solutions, the resulting approximation error remains bounded and does not interfere with the contraction property of $\mathcal{P}\mathcal{T}^\pi$, allowing the learned critic to track the fixed point stably.

### 3.3 POLICY UPDATE WITH SCVAR

After obtaining the GMM parameters using the safety critic, we use $\text{SCVaR}_\alpha$ to guide safe exploration and improve policy optimization. At each iteration, the GMM parameters $\{(\mu_k, \sigma_k, \omega_k)\}_{k=1}^K$ are estimated, enabling the computation of the safety measure $\Lambda^\pi(s, a, \alpha)$ for a specified risk level $\alpha$:

$$\Lambda^\pi(s, a, \alpha) \doteq \text{SCVaR}_\alpha \left( \{(\mu_k, \sigma_k, \omega_k)\}_{k=1}^K \right). \tag{17}$$

The policy is optimized under the constraint:

$$\Lambda^\pi(s, a, \alpha) \leq d, \quad \forall t, \tag{18}$$

where $d$ is a discounted safety threshold derived from the episodic constraint $D$. The threshold is given by

$$d = \frac{D \cdot \left(1 - \gamma^T\right)}{(1 - \gamma) \cdot T}, \tag{19}$$

where $\gamma$ is the discount factor and $T$ is the maximum episode length. This per-step formulation is not intended to impose a stricter constraint. It provides a discount-consistent normalization that ensures the cumulative per-step cost matches the original episodic limit $D$. Such normalization is necessary because SCVaR is defined on one-step safety critics, and without it an agent might satisfy all per-step risks while still violating the episode-level bound. This setting is also consistent with WC-SAC (Yang et al., 2021) and its official implementation. The detailed derivation is provided in Appendix D.4.

To balance task performance and safety, we extend the SAC framework (Haarnoja et al., 2018) by introducing a safety-adjusted target distribution. Specifically, the policy $\pi_\theta$ is optimized by minimizing the KL divergence between the current policy and a target distribution that incorporates both reward maximization and safety regularization:

$$\min_\pi D_{\mathrm{KL}} \left( \pi(\cdot \mid s_t) \middle\| \frac{\exp\left(\frac{1}{\lambda}\left(Q_r^\pi(s_t, \cdot) - \kappa \Lambda^\pi(s_t, \cdot, \alpha)\right)\right)}{Z^\pi(s_t)} \right), \tag{20}$$

where $Z^\pi(s_t)$ is the partition function ensuring normalization, $\lambda > 0$ is the temperature parameter controlling entropy, and $\kappa > 0$ is the safety weight trading off rewards against constraint violations.

From this formulation, we derive the actor loss function:

$$J_\pi(\theta) = \mathbb{E}_{(s_t, a_t) \sim \rho_{\pi_\theta}} \left[ \lambda \log \pi_\theta(a_t \mid s_t) - X_{\alpha,\kappa}^{\pi_\theta}(s_t, a_t) \right], \tag{21}$$

where $X_{\alpha,\kappa}^{\pi_\theta}(s, a) = Q_r^\pi(s, a) - \kappa \Lambda^\pi(s, a, \alpha)$, with $Q_r^\pi(s, a)$ denoting the standard state-action value function.

To further guarantee adherence to safety constraints, the safety weight $\kappa$ is not fixed but adaptively tuned. This is achieved by minimizing

$$J_s(\kappa) = \mathbb{E}_{(s_t, a_t) \sim \rho_{\pi_\theta}} \left[ \kappa \left( d - \Lambda^\pi(s_t, a_t, \alpha) \right) \right]. \tag{22}$$

### 3.4 Overall Framework

The overall architecture of our GMM SCVaR-Guided SAC (GMM-SSAC) is illustrated in Fig. 3. It augments the SAC framework with a GMM-based distributional safety critic and an SCVaR-constrained policy update.

The environment generates transitions $(s, a, s', r, c)$, which are stored in the replay buffer and used to update three components:

**(1) GMM-Based Safety Critic.** For sampled transitions, we (a) draw next-state actions, (b) compute the Bellman distribution, (c) blend historical and Bellman-transformed samples via the refinement operator, and (d) apply an EM projection onto the GMM. The online critic then regresses toward the EM-updated parameters and is softly aligned with the target critic.

**(2) Reward Critic.** This branch follows standard SAC with twin Q-functions, a MaxEnt Bellman update, and soft target updates.

---

**Algorithm 1** GMM-SSAC

1: Initialize policy $\pi_\theta$, reward critics $(Q_1, Q_2)$, GMM safety critic $\mathcal{G}^\pi$, target nets, dual weight $\kappa$, replay buffer $\mathcal{D}$.
2: **while** not converged **do**
3:     **Interact**: collect $(s, a, s', r, c)$ and store in $\mathcal{D}$.
4:     **Safety critic**: sample mini-batch; draw $a' \sim \pi_\theta(\cdot|s')$; form Bellman samples $\Psi_B$; refine $\Psi_{\mathrm{update}} = \mathcal{R}(\Psi, \Psi_B, \beta)$; EM fit GMM $(\mu_k, \sigma_k, \omega_k)$; MSE regress $\mathcal{F}_\psi$ and soft-update target.
5:     **Reward critic**: SAC double-Q Bellman update, soft-update targets.
6:     **Risk and policy**: compute $\mathrm{SCVaR}_\alpha$ from GMM; update $\pi_\theta$ with primal–dual objective using reward and $\mathrm{SCVaR}_\alpha$.
7:     **Dual**: update $\kappa$ to enforce $\mathrm{SCVaR}_\alpha \leq d$.
8: **end while**
9: **return** $\pi_\theta$.

---

**(3) Policy Update.** The policy receives reward signals from the Q-functions and risk signals from the GMM safety critic. SCVaR is computed analytically from GMM components, and the actor is optimized via a primal–dual objective to balance reward and safety. The dual weight $\kappa$ is adapted online to enforce the SCVaR threshold ($d$).

A concise overview of the training loop is provided in Alg. 1, with full details in Appendix C.

## 4 Experiments

We conduct a comparative evaluation using CarGoal1, CarButton1, and CarCircle1 from the Safety-Gymnasium benchmark [1] (Ji et al., 2023) and Hopper and Ant from the velocity-constrained MuJoCo benchmark [2] (Todorov et al., 2012). A comprehensive description of the tested tasks and hyper-parameters is provided in Appendix D.

---

[1]https://github.com/PKU-Alignment/safety-gymnasium
[2]https://github.com/google-deepmind/mujoco

The baselines considered in our experiments are as follows: (1) SAC (Haarnoja et al., 2018), a method without safety constraints, allowing us to analyze the reward-cost trade-offs in each experimental environment; (2) SAC-Lag (Stooke et al., 2020), which employs a Lagrange multiplier update method that leverages the derivatives of the safety constraint; (3) WC-SAC (Yang et al., 2021), which uses a Gaussian distribution and CVaR estimation to model the safety constraint; and (4) CAL (Wu et al., 2024), which models the safety constraint using multiple Gaussian distributions and derives an UCB by aggregating multiple independent cost distribution estimates.

In addition to the baseline methods, we categorize the GMM-SSAC method into three variants based on different risk level values $\alpha$ for SCVaR: GMM-0.1 ($\alpha = 0.1$, the most conservative), GMM-0.5 ($\alpha = 0.5$, moderate risk aversion), and GMM-0.9 ($\alpha = 0.9$, nearly disregarding risk). The implementation detail and hyperparameter settings of all models are given in Appendix D.2.

We conducted several ablation studies to assess the robustness and interpretability of our approach. In the main text, we highlight two key experiments: (i) varying the number of Gaussian components and (ii) tuning the sample–set blending ratio $\beta$. Additional ablations, including comparisons with alternative risk measures, integration into baseline methods, and component-level interpretability analyses, are provided in the Appendix E.

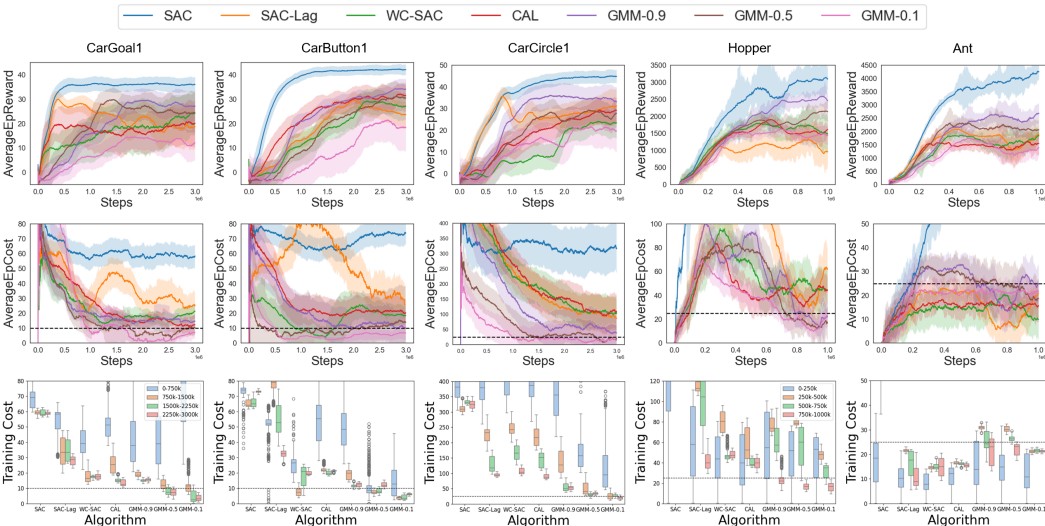

Figure 4: Comparison with off-policy baselines on five benchmark environments. Top/middle rows show average rewards and costs, bottom row shows training costs (divided into four phases). GMM-SSAC consistently reduces safety violations while maintaining competitive rewards, with $\alpha$ controlling the reward–cost trade-off. In Ant and Hopper, SCVaR adapts to the intrinsic reward–cost coupling, achieving balanced performance.

### 4.1 MAIN RESULTS

Fig. 4 presents the benchmark results averaged over 5 random seeds, demonstrating the effectiveness of the GMM-based approach compared to conventional Gaussian-based methods. The top and middle rows report reward and cost evaluated using additional test episodes after each training iteration. The bottom row visualizes the training-time cost, where training episodes are divided into four equal segments and shown as boxplots for each algorithm.

Across CarGoal1, CarButton1, CarCircle1, and Hopper, GMM-SSAC with $\alpha = 0.1$ and $\alpha = 0.5$ consistently reduces safety violations after 750k steps while achieving comparable or superior reward and constraint satisfaction relative to state-of-the-art off-policy methods. By tuning $\alpha$, the algorithm can navigate the reward–cost trade-off: with $\alpha = 0.9$, the model approaches the reward performance of unconstrained SAC while eventually reducing costs to acceptable levels due to the worst-case sensitivity of SCVaR.

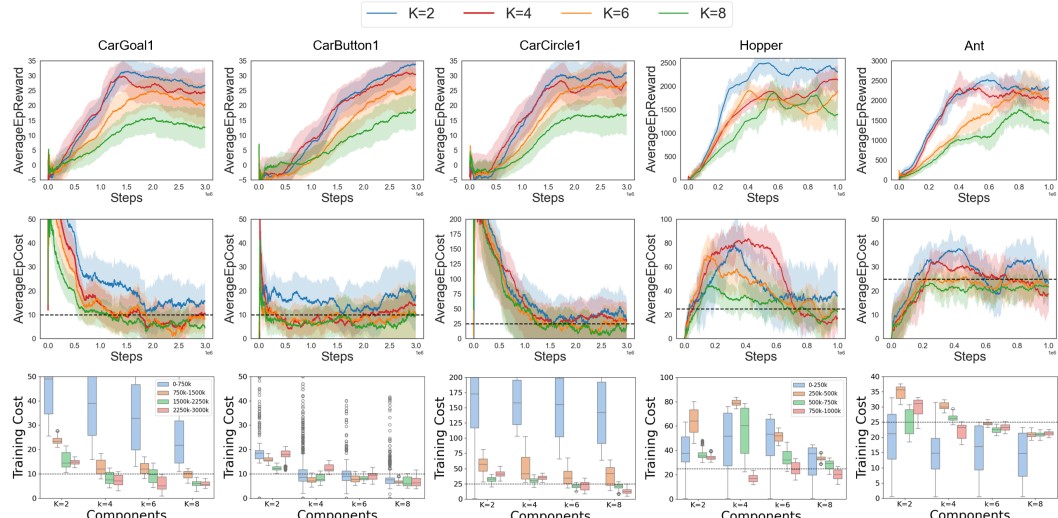

Figure 5: Ablation on the number of GMM components ($K = 2, 4, 6, 8$). Fewer components yield higher rewards but weaker cost control, while larger $K$ improves safety by better modeling heavy-tailed distributions. Results show that $K = 6, 8$ achieve lower and more stable training costs across environments, especially in CarGoal1, CarButton1, and CarCircle1.

In the Ant environment, although our model maintains acceptable costs and achieves higher rewards compared to baselines, the cost is slightly higher because the reward and safety metrics are tightly coupled through velocity. Surpassing the velocity threshold yields penalties, whereas higher velocities also yield higher rewards. GMM-SSAC iteratively adjusts the mixture distribution so that SCVaR tracks the safety limit $d$ closely without surpassing it, which is the sense in which our method achieves a precise safety–reward balance in this environment.

Since different tasks exhibit different safety-cost distributions, the most suitable value of $\alpha$ naturally varies across environments. This is also reflected in our results: $\alpha = 0.5$ often leads to a more favorable reward–cost compromise, whereas $\alpha = 0.1$ emphasizes stricter constraint satisfaction.

## 4.2 ABLATION STUDY

All ablation experiments are conducted with a fixed risk level $\alpha = 0.5$, which provides a balanced degree of risk sensitivity across different tasks.

### 4.2.1 NUMBER OF GAUSSIAN COMPONENTS

Fig. 5 compares the performance of models using 2, 4, 6, and 8 GMM components. The results demonstrate a clear trade-off between reward performance and safety constraints across environments. While models with fewer components ($K = 2, 4$) achieve higher rewards, configurations with more components ($K = 6, 8$) consistently maintain lower costs during training. This is particularly evident in CarGoal1, CarButton1, and CarCircle1, where $K = 8$ achieves the lowest safety violations. In the Hopper environment, $K = 6$ and $K = 8$ demonstrate more stable cost control compared to $K = 2$, which shows higher variance in constraint satisfaction. The bottom row statistics further confirm this pattern, showing that models with more components generally maintain lower training costs across different stages, especially in the later phases (750k-3000k steps). Given the complex nature of cumulative cost value distributions, incorporating additional components enables more accurate modeling of heavy-tailed values. This improved representation of tail distributions leads to more precise estimation of worst-case scenarios, thereby enhancing the model's conservative behavior and safety guarantees.

While increasing the number of components improves the ability of the GMM critic to capture multi-modality and heavy-tailed structure, an excessively large $K$ may introduce additional variance during learning. This is because (i) the EM update becomes more sensitive to sampling noise when responsibilities are distributed across many components, and (ii) the Bellman-transformed sample

set $\Psi_B(s, a)$ inherently contains randomness from both the policy and the environment. When $K$ is too large, these sources of variance compound, leading to occasional fluctuations in component weights and means, which in turn affect SCVaR estimates. Nevertheless, we find that moderate values ($K=4$–$6$) strike a good balance between expressiveness and stability across all tasks, and the contraction property of the Bellman–EM operator ensures that the model does not diverge even in these higher-dimensional mixture settings.

### 4.2.2 SAMPLE-SET BLENDING RATIO

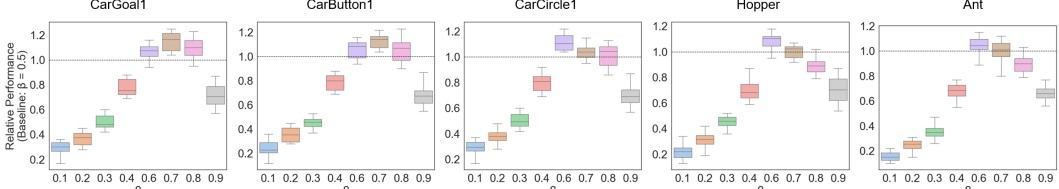

Figure 6: Ablation Study on the Impact of Blending Ratio $\beta$.

To examine the impact of the blending ratio $\beta$ in the incremental updating process, we use $\beta = 0.5$ as the baseline. The relative performance is defined as:

$$P_R(x) = \frac{1}{2} \times \frac{R_{\beta=x}}{R_{\beta=0.5}} + \frac{1}{2} \times \frac{C_{\beta=0.5}}{C_{\beta=x}}, \tag{23}$$

where $R_{\beta=x}$ and $C_{\beta=x}$ are the reward and cost values under a specific $\beta$. This metric reflects the balance between rewards and costs for different $\beta$ settings.

Fig. 6 shows the performance comparison of various blending ratios $\beta$ during incremental updates, based on results averaged over ten runs with the same random seed in the same environment. For Safety-Gymnasium, the optimal $\beta$ lies between 0.5 and 0.8, while for Mujoco, it is between 0.5 and 0.7. The optimal range highlights the importance of weighting newer learning targets more heavily in RL to avoid early stagnation due to the dynamic nature of the targets. We observe that setting $\beta = 0.9$ yields suboptimal performance. This can be attributed to the variance introduced in GMM parameters when using EM updates alongside Bellman equation updates.

These results further reveal that the blending ratio $\beta$ plays a crucial role in stabilizing the Bellman–EM update. A moderate value of $\beta$ allows the update to exploit fresh Bellman targets while still smoothing them with historical samples, effectively reducing variance. However, very large $\beta$ ($\geq$ 0.9) causes the update to rely almost exclusively on newly generated $\Psi_B(s, a)$, which amplifies noise in both the Bellman estimate and the subsequent EM fitting step. This overemphasis on noisy, non-stationary targets leads to fluctuations in component assignments and tail estimates, degrading SCVaR stability. On the other hand, overly small $\beta$ slows learning by anchoring the critic too strongly to past distributions, reducing its ability to track evolving policies. Overall, the empirical ranges identified in Safety-Gymnasium and Mujoco reflect the trade-off between adaptivity and variance control that is intrinsic to incremental mixture-model updates.

## 5 CONCLUSION

We present GMM-SSAC, a Safe RL framework that combines GMMs with SCVaR-based risk assessment to improve safety in complex environments. Experiments show that GMM-SSAC achieves stronger safety guarantees while maintaining competitive reward performance. Key findings include: (1) GMM-based safety critics better capture complex risk distributions compared to single Gaussian approaches; (2) SCVaR effectively manages worst-case risks by considering the maximum CVaR across mixture components; and (3) the incremental updating mechanism enables stable learning in dynamic environments. These results suggest that more expressive risk modeling through mixture distributions, combined with conservative risk measures, provides a promising direction for developing robust safe RL systems. Future work could explore adaptive component selection strategies and extend the framework to handle more complex safety constraints.

## ETHICS STATEMENT

This work adheres to the ICLR Code of Ethics.[3] Our research focuses on methodological advances in safe reinforcement learning and does not involve human subjects, personally identifiable information, or sensitive user data. The experiments are conducted entirely in simulated environments, ensuring that no harm is caused to humans, animals, or the environment.

The proposed methods aim to improve the safety and reliability of reinforcement learning algorithms, particularly in high-stakes applications. While our approach provides a more conservative mechanism for risk-sensitive decision making, it does not directly enable harmful applications. We emphasize that any future deployment in real-world safety-critical domains (e.g., healthcare, autonomous driving, finance) must carefully account for regulatory, ethical, and societal considerations beyond the scope of this work.

We disclose no conflicts of interest, external sponsorship, or ethical concerns related to fairness, discrimination, privacy, or security. All experiments were designed and conducted in compliance with established standards of research integrity.

## REPRODUCIBILITY STATEMENT

We have made significant efforts to ensure the reproducibility of our work. All theoretical results are presented with complete proofs in the appendix, and detailed algorithmic descriptions, including pseudocode, are provided in Section 3 and Appendix C. Experimental setups, hyperparameters, and evaluation protocols are fully described in Section 4 and Appendix D. In addition, we will release our anonymized implementation in the supplementary materials to facilitate independent verification of the reported results.

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

TABLE OF CONTENTS

# A  SCVaR ADVANTAGE ANALYSIS

## A.1  THEORETICAL PROOF: SCVaR EXCEEDS CVaR IN GMMs

*Proof.* Assume the loss variable $L$ follows a GMM with $K$ components:

$$f_L(x) = \sum_{i=1}^{K} w_i \cdot \phi_i(x), \tag{24}$$

where $w_i$ are the mixture weights ($\sum w_i = 1$), and $\phi_i(x) = \mathcal{N}(x \mid \mu_i, \sigma_i^2)$ is the PDF of the $i$-th Gaussian component.

The VaR at confidence level $\alpha$ is defined as the $(1 - \alpha)$-quantile:

$$\text{VaR}_\alpha(L) = \inf\{q \mid \Pr[L \geq q] \leq 1 - \alpha\} = \text{ICDF}_L(1 - \alpha),$$

which for a GMM satisfies:

$$\sum_{i=1}^{K} w_i \cdot \Phi\left(\frac{\text{VaR}_\alpha - \mu_i}{\sigma_i}\right) = 1 - \alpha,$$

where $\Phi(\cdot)$ is the standard normal CDF.

The Conditional Value-at-Risk (CVaR) is defined as the expected loss conditional on exceeding VaR:

$$\text{CVaR}_\alpha(L) = \mathbb{E}[L \mid L \geq \text{VaR}_\alpha] = \frac{1}{1 - \alpha} \int_{\text{VaR}_\alpha}^{\infty} x \cdot f_L(x) \, dx.$$

Expanding over the mixture:

$$\text{CVaR}_\alpha^{\text{GMM}} = \frac{1}{1 - \alpha} \sum_{i=1}^{K} w_i \int_{\text{VaR}_\alpha}^{\infty} x \cdot \phi_i(x) \, dx.$$

For each Gaussian component, we apply the change of variables:

$$z_i = \frac{x - \mu_i}{\sigma_i}, \quad t = \mu_i + \sigma_i z_i, \quad dt = \sigma_i \, dz_i.$$

This transforms the integral:

$$\int_{\text{VaR}_\alpha}^{\infty} t \cdot \phi_i(t) \, dt = \int_{z_i}^{\infty} (\mu_i + \sigma_i z) \cdot \phi(z) \, dz, \quad z_i = \frac{\text{VaR}_\alpha - \mu_i}{\sigma_i}.$$

Splitting the terms, we get:

$$\mu_i \int_{z_i}^{\infty} \phi(z) \, dz + \sigma_i \int_{z_i}^{\infty} z \cdot \phi(z) \, dz.$$

These evaluate as:

$$\int_{z_i}^{\infty} \phi(z) \, dz = 1 - \Phi(z_i), \quad \int_{z_i}^{\infty} z \cdot \phi(z) \, dz = \phi(z_i),$$

yielding:

$$\int_{\text{VaR}_\alpha}^{\infty} t \cdot \phi_i(t) \, dt = \mu_i(1 - \Phi(z_i)) + \sigma_i \phi(z_i).$$

Therefore, the overall CVaR is:

$$\text{CVaR}_\alpha^{\text{GMM}} = \frac{1}{1 - \alpha} \sum_{i=1}^{K} w_i \left[\mu_i(1 - \Phi(z_i)) + \sigma_i \phi(z_i)\right].$$

To express this as a convex combination, define the posterior tail weights:

$$\text{PostWeight}_i = \frac{w_i(1 - \Phi(z_i))}{1 - \alpha},$$

which satisfy $\sum_{i=1}^{K} \text{PostWeight}_i = 1$.

Thus, we can write:

$$\text{CVaR}_\alpha^{\text{GMM}} = \sum_{i=1}^{K} \text{PostWeight}_i \cdot \text{CVaR}_\alpha^{(i)}.$$

We define the SCVaR as:

$$\text{SCVaR}_\alpha = \max_i \text{CVaR}_\alpha^{(i)},$$

which captures the worst-case per-component tail risk (based only on its own distribution).

Since $\text{CVaR}_\alpha^{\text{GMM}}$ is a convex combination of per-component contributions, it holds that:

$$\sum p_i a_i \leq \max_i a_i \quad \text{for} \quad p_i \geq 0, \sum p_i = 1,$$

implying:

$$\text{CVaR}_\alpha^{\text{GMM}} \leq \text{SCVaR}_\alpha.$$

Therefore, SCVaR always provides a conservative upper bound on the mixture CVaR, explicitly focusing on the worst-case per-component tail risk.

This completes the proof.

$\square$

## A.2 DETAILED PROOF OF SCVAR COHERENCE

*Proof.* Let the (cost) distribution be modeled by a Gaussian mixture. For

$$X = \sum_{k=1}^{K} \omega_k \, \mathcal{N}(\mu_k, \sigma_k^2), \qquad \sum_k \omega_k = 1, \; \omega_k \geq 0,$$

define

$$\text{SCVaR}_\alpha(X) := \sup_{k=1,\dots,K} \text{CVaR}_\alpha^{(k)}(X), \qquad \text{CVaR}_\alpha\big(\mathcal{N}(\mu, \sigma^2)\big) = \mu + c_\alpha \, \sigma, \quad c_\alpha := \frac{\phi\big(\Phi^{-1}(1 - \alpha)\big)}{1 - \alpha}.$$

Note that $\text{CVaR}_\alpha^{(k)}(X)$ depends only on the component parameters $(\mu_k, \sigma_k)$, not on the mixture weight.

**Monotonicity.** If $X \leq Y$ almost surely, then for all $k$, $\text{CVaR}_\alpha^{(k)}(X) \leq \text{CVaR}_\alpha^{(k)}(Y)$ (monotonicity of CVaR). Taking the supremum over $k$ yields

$$\text{SCVaR}_\alpha(X) = \sup_k \text{CVaR}_\alpha^{(k)}(X) \leq \sup_k \text{CVaR}_\alpha^{(k)}(Y) = \text{SCVaR}_\alpha(Y).$$

**Translation invariance.** For any constant $a \in \mathbb{R}$,

$$\text{CVaR}_\alpha^{(k)}(X + a) = \text{CVaR}_\alpha^{(k)}(X) + a \quad \Rightarrow \quad \text{SCVaR}_\alpha(X + a) = \text{SCVaR}_\alpha(X) + a.$$

**Positive homogeneity.** For any $\lambda \geq 0$,

$$\text{CVaR}_\alpha^{(k)}(\lambda X) = \lambda \, \text{CVaR}_\alpha^{(k)}(X) \quad \Rightarrow \quad \text{SCVaR}_\alpha(\lambda X) = \lambda \, \text{SCVaR}_\alpha(X).$$

**Subadditivity.** Let

$$X = \sum_{k=1}^{K} \omega_k \, \mathcal{N}(\mu_k, \sigma_k^2), \qquad Y = \sum_{l=1}^{L} \lambda_l \, \mathcal{N}(\nu_l, \tau_l^2), \qquad \sum_k \omega_k = \sum_l \lambda_l = 1, \ \omega_k, \lambda_l \geq 0,$$

and set $Z := X + Y$. Each pair $(k, l)$ induces a Gaussian component of $Z$ with

$$\mu_{k,l} = \mu_k + \nu_l, \qquad \sigma_{k,l} = \sqrt{\sigma_k^2 + \tau_l^2}, \qquad \text{weight } \omega_k \lambda_l.$$

Using the closed form of Gaussian CVaR,

$$\mathrm{CVaR}_\alpha^{(k,l)}(Z) = (\mu_k + \nu_l) + c_\alpha \sqrt{\sigma_k^2 + \tau_l^2} \ \leq \ (\mu_k + c_\alpha \sigma_k) + (\nu_l + c_\alpha \tau_l) = \mathrm{CVaR}_\alpha^{(k)}(X) + \mathrm{CVaR}_\alpha^{(l)}(Y),$$

where $\sqrt{a^2 + b^2} \leq a + b$ for $a, b \geq 0$. Taking the supremum and using $\sup_{(k,l)}(a_k + b_l) \leq \sup_k a_k + \sup_l b_l$,

$$\mathrm{SCVaR}_\alpha(Z) = \sup_{(k,l)} \mathrm{CVaR}_\alpha^{(k,l)}(Z) \ \leq \ \sup_{(k,l)} \left( \mathrm{CVaR}_\alpha^{(k)}(X) + \mathrm{CVaR}_\alpha^{(l)}(Y) \right)$$

$$\leq \ \sup_k \mathrm{CVaR}_\alpha^{(k)}(X) + \sup_l \mathrm{CVaR}_\alpha^{(l)}(Y) = \mathrm{SCVaR}_\alpha(X) + \mathrm{SCVaR}_\alpha(Y).$$

Therefore, SCVaR satisfies monotonicity, translation invariance, positive homogeneity, and subadditivity, and is thus a coherent risk measure. □

### A.3 Investment Portfolio Example Involving Bonds and Stocks

Consider a portfolio modeled by a two-component Gaussian Mixture Model (GMM), where each component represents a distinct investment class:

$$f(x) = \omega_1 \mathcal{N}(x; \mu_1, \sigma_1^2) + \omega_2 \mathcal{N}(x; \mu_2, \sigma_2^2).$$

where the first component represents stocks with parameters:

$$\mu_1 = 10, \quad \sigma_1 = 20, \quad \omega_1 = 0.6,$$

and the second component represents bonds with parameters:

$$\mu_2 = 5, \quad \sigma_2 = 5, \quad \omega_2 = 0.4.$$

The mixture weights indicate a 60% probability to stocks and 40% to bonds. For a confidence level $\alpha = 0.95$, we employ Monte-Carlo sampling to compute both CVaR and SCVaR for this portfolio. The results are:

$$\mathrm{CVaR}_{0.95} = 29.1713, \quad \mathrm{SCVaR}_{0.95} = \mathrm{CVaR}_{0.95}^{\mathrm{Stock}} = 30.3136.$$

In this example, SCVaR accounts for the inherent possibility that **an investor may concentrate investments in stocks**. By recognizing that the maximum risk exposure stems from the stock component's inherent volatility, SCVaR provides a more conservative risk estimate compared to the standard CVaR.

Given that the parameters $\{\mu_1, \mu_2, \sigma_1, \sigma_2, \omega_1, \omega_2\}$ are outputs of a neural network, the safety measure as a function of $\omega_k$ within the interval $(0, 1]$ satisfies the following two properties:

**Theorem 3.** *For a given risk level $\alpha$, the safety measure $\Lambda^\pi(s, a, \alpha) = SCVaR_\alpha$ is **invariant to** $\omega_k \in (0, 1]$ for each $k \in \{1, 2, \cdots, K\}$. Specifically,*

$$\frac{\partial \Lambda^\pi(s, a, \alpha)}{\partial \omega_k} = 0, \quad for \quad k \in \{1, 2, \cdots, K\}, \quad \omega_k \in (0, 1].$$

**Theorem 4.** *For a given risk level $\alpha$, the safety measure $\Lambda^\pi(s, a, \alpha) = SCVaR_\alpha$ is **invariant to** $\mu_k$ and $\sigma_k$ for all $k \in \{1, 2, \ldots, K\}$ except for the $k_{max}$-th components associated with the largest CVaR. Specifically,*

$$\frac{\partial \Lambda^\pi(s, a, \alpha)}{\partial \mu_k} = 0 \quad and \quad \frac{\partial \Lambda^\pi(s, a, \alpha)}{\partial \sigma_k} = 0, \quad \forall k \in \{1, 2, \ldots, K\} \setminus \{k_{max}\},$$

*where $k_{max}$ is the index of the component associated with the largest CVaR, i.e.,*

$$k_{max} = \arg \max_{k \in \{1, 2, \ldots, K\}} CVaR_\alpha^{(k)}.$$

Through the above two theorems, we know that under the guidance of SCVaR, the RL agent tends to completely exclude components that may lead to higher risks. In this case, if a stock is considered a high-risk investment, the RL agent will completely avoid this option, or SCVaR will mitigate the risk associated with the stock choice (for example, by selecting another stock with relatively lower return variance). If an investor completely abandons stocks and chooses a less risky investment compared to bonds, SCVaR would encourage the investor to entirely forgo the bond option as well.

## B  INCREMENTAL UPDATING FOR SAFETY CRITIC WITH BELLMAN OPERATOR AND ITS CONVERGENCE

---

**Algorithm 2** Incremental Updating for Safety Critic with Bellman Operator

---

1: **Input:** Current GMM parameters $\Gamma^\pi(s,a) = \{(\mu_k, \sigma_k, \omega_k)\}_{k=1}^K$, Immediate cost $c(s,a)$, Discount factor $\gamma$, Total samples $M$, Blending coefficient $\beta$, Neural network $\mathcal{F}$ for predicting $\Gamma^\pi(s,a)$.

2: **Output:** Updated network parameters for $\mathcal{F}$.

3: **Initialize:** Generate samples $\Psi(s,a) = \mathcal{X}(\mathcal{G}^\pi(s,a), M)$ using the current GMM.

4: Generate samples $\Psi(s',a') = \mathcal{X}(\mathcal{G}^\pi_{\text{target}}(s',a'), M)$ using the target safety critic GMM.

   **Step 1: Bellman Sampling**

5: Transform samples using the Bellman operator:

$$\Psi_{\mathcal{B}}(s,a) = \{\hat{x}_i \mid \hat{x}_i = c(s,a) + \gamma x_i', \ x_i' \in \mathcal{X}(\mathcal{G}^\pi_{\text{target}}(s',a'), M)\}.$$

   **Step 2: Incremental Refinement**

6: Blend current and Bellman-transformed samples:

$$\Psi_{\text{update}}(s,a) = \mathcal{R}(\Psi(s,a), \Psi_{\mathcal{B}}(s,a), \beta) = \{x_i, ..., x_{M_1}, c(s,a)+\gamma x_1', ..., c(s,a)+\gamma x_{M_2}' | M_1 : M_2 = (1-\beta) : \beta\}$$

   **Step 3: Projection Operation**

7: Perform EM to update GMM parameters:

8: **E-step:** :

$$\gamma_{mk} = \frac{\omega_k \varphi(x_m \mid \mu_k, (\sigma_k)^2)}{\sum_{j=1}^K \omega_j \varphi(x_m \mid \mu_j, (\sigma_j)^2)}.$$

9: **M-step:** :

$$\mu_k^{\text{update}} = \frac{\sum_{m=1}^M \gamma_{mk} x_m}{\sum_{m=1}^M \gamma_{mk}}, \quad (\sigma_k^{\text{update}})^2 = \frac{\sum_{m=1}^M \gamma_{mk}(x_m - \mu_k^{\text{update}})^2}{\sum_{m=1}^M \gamma_{mk}}, \quad \omega_k^{\text{update}} = \frac{\sum_{m=1}^M \gamma_{mk}}{M}.$$

10: Normalize mixing coefficients:

$$\omega_k^{\text{update}} \leftarrow \frac{\omega_k^{\text{update}}}{\sum_{j=1}^K \omega_j^{\text{update}}}, \quad \forall k \in \{1, \ldots, K\}.$$

   **Step 4: Neural Network Update**

11: Compute the Mean Squared Error (MSE) loss between predicted GMM parameters $(\mu_k, \sigma_k, \omega_k)$ and update parameters $(\mu_k^{\text{update}}, \sigma_k^{\text{update}}, \omega_k^{\text{update}})$:

$$\mathcal{L} = \sum_{k=1}^K \left[ (\mu_k - \mu_k^{\text{update}})^2 + (\sigma_k - \sigma_k^{\text{update}})^2 + (\omega_k - \omega_k^{\text{update}})^2 \right].$$

12: Perform gradient descent on the network parameters of $\mathcal{F}$ to minimize $\mathcal{L}$.

13: **Return:** Updated GMM parameters $\Gamma^\pi_{\text{update}}(s,a)$ and updated network $\mathcal{F}$.

---

*Proof.* **Setup.** Fix a policy $\pi$. Let $\nu(\cdot \mid s, a) \in \mathcal{P}_1(\mathbb{R})$ denote the (cost) value distribution at $(s, a)$. Project $\nu(\cdot \mid s, a)$ onto a Gaussian mixture:

$$\mathcal{V}(s, a) = \sum_{k=1}^{K} \omega_k \, \mathcal{N}(x \mid \mu_k, \sigma_k), \qquad \omega_k \geq 0, \ \sum_{k=1}^{K} \omega_k = 1.$$

The distributional Bellman operator (cost form) is

$$(\mathcal{T}^\pi \nu)(\cdot \mid s, a) = \mathsf{Law}(c(s, a) + \gamma \, X'), \quad X' \sim \nu(\cdot \mid s', a'), \ (s', a') \sim P(\cdot \mid s, a) \times \pi(\cdot \mid s').$$

One SC-MGR update is the Bellman–EM composition

$$\mathcal{B}^\pi \nu \ \equiv \ \mathcal{P}\big(\mathcal{T}^\pi \nu\big),$$

where $\mathcal{P}$ denotes the EM projection onto the GMM family.

**Bellman contraction under** $W_1$. By the Kantorovich–Rubinstein duality,

$$W_1(\mu, \nu) = \sup_{\|f\|_{\mathrm{Lip}} \leq 1} \Big| \mathbb{E}_\mu[f] - \mathbb{E}_\nu[f] \Big|.$$

Two basic invariances of $W_1$ (for any constant $b$ and any $a \geq 0$) are

$$W_1(\mathsf{Law}(X + b), \mathsf{Law}(Y + b)) = W_1(\mathsf{Law}(X), \mathsf{Law}(Y)),$$

$$W_1(\mathsf{Law}(aX), \mathsf{Law}(aY)) = a \, W_1(\mathsf{Law}(X), \mathsf{Law}(Y)).$$

Fix $(s, a)$ and two distributions $\nu_1, \nu_2$. Conditioning on $(s', a')$ and applying the two invariances gives

$$\begin{aligned}
W_1\big((\mathcal{T}^\pi \nu_1)(\cdot \mid s, a), (\mathcal{T}^\pi \nu_2)(\cdot \mid s, a)\big) &= W_1\big(\mathsf{Law}(c + \gamma X_1'), \mathsf{Law}(c + \gamma X_2')\big) \\
&= \gamma \, W_1\big(\mathsf{Law}(X_1'), \mathsf{Law}(X_2')\big) \\
&\leq \gamma \, \mathbb{E}_{(s', a')}\Big[ W_1\big(\nu_1(\cdot \mid s', a'), \nu_2(\cdot \mid s', a')\big) \Big] \\
&\leq \gamma \sup_{(s', a')} W_1\big(\nu_1(\cdot \mid s', a'), \nu_2(\cdot \mid s', a')\big).
\end{aligned}$$

Hence, uniformly over $(s, a)$,

$$W_1\big(\mathcal{T}^\pi \nu_1, \mathcal{T}^\pi \nu_2\big) \ \leq \ \gamma \, W_1(\nu_1, \nu_2), \qquad \gamma \in (0, 1).$$

**EM projection: responsibilities and unbiased M-step.** Given samples $\{x_m\}_{m=1}^{M} \sim p(x)$ (here $p = \mathcal{T}^\pi \nu$), the E-step responsibility for component $k$ is

$$\gamma_{mk} \ = \ \frac{\omega_k \, \varphi(x_m \mid \mu_k, \sigma_k^2)}{\sum_{j=1}^{K} \omega_j \, \varphi(x_m \mid \mu_j, \sigma_j^2)}, \qquad \varphi(x \mid \mu, \sigma^2) = \frac{1}{\sqrt{2\pi}\sigma} \exp\Big(-\frac{(x - \mu)^2}{2\sigma^2}\Big).$$

In the population limit ($M \to \infty$), the M-step updates are

$$\mu_k^{\mathrm{new}} = \frac{\mathbb{E}_p[\gamma_k(X) \, X]}{\mathbb{E}_p[\gamma_k(X)]}, \qquad \omega_k^{\mathrm{new}} = \mathbb{E}_p[\gamma_k(X)].$$

If $p$ equals the current GMM (or is at an EM fixed point), i.e., $p(x) = \sum_j \omega_j \varphi(x \mid \mu_j, \sigma_j^2)$, then

$$\mathbb{E}_p[\gamma_k(X)] = \int \frac{\omega_k \varphi(x \mid \mu_k, \sigma_k^2)}{\sum_j \omega_j \varphi(x \mid \mu_j, \sigma_j^2)} \, p(x) \, dx = \int \omega_k \varphi(x \mid \mu_k, \sigma_k^2) \, dx = \omega_k,$$

and

$$\mathbb{E}_p[\gamma_k(X) \, X] = \int \omega_k \varphi(x \mid \mu_k, \sigma_k^2) \, x \, dx = \omega_k \, \mu_k.$$

Therefore,

$$\mathbb{E}[\mu_k^{\mathrm{new}}] = \frac{\omega_k \, \mu_k}{\omega_k} = \mu_k, \qquad \mathbb{E}[\omega_k^{\mathrm{new}}] = \omega_k,$$

i.e., the EM projection preserves component means and weights in expectation (population unbiasedness), hence does not distort first-moment structure.

**Bellman–EM composition and convergence.** Let $\nu_{t+1} = \mathcal{P}\mathcal{T}^\pi \nu_t$. Using the Bellman contraction above and the non-expansiveness of $\mathcal{P}$ in $W_1$ (i.e., $W_1(\mathcal{P}\mu, \mathcal{P}\nu) \leq W_1(\mu, \nu)$),

$$W_1(\nu_{t+1}, \nu'_{t+1}) = W_1\big(\mathcal{P}\mathcal{T}^\pi \nu_t, \ \mathcal{P}\mathcal{T}^\pi \nu'_t\big) \ \leq \ W_1\big(\mathcal{T}^\pi \nu_t, \ \mathcal{T}^\pi \nu'_t\big)$$
$$\leq \ \gamma \, W_1(\nu_t, \nu'_t).$$

Thus $\mathcal{P}\mathcal{T}^\pi$ is a $\gamma$-contraction under $W_1$. By the Banach fixed-point theorem, there exists a unique $\mathcal{V}^*$ such that

$$\mathcal{P}\mathcal{T}^\pi \mathcal{V}^* = \mathcal{V}^*, \qquad W_1(\nu_t, \mathcal{V}^*) \ \leq \ \gamma^t \, W_1(\nu_0, \mathcal{V}^*).$$

Together with the population unbiasedness above, the Bellman–EM updates of SC-MGR converge stably to the unique distributional fixed point $\mathcal{V}^*$. □

## C   DETAIL DERIVATION OF POLICY UPDATE & COMPLETE ALGORITHM: GMM-SSAC

### C.1   DETAIL DERIVATION OF POLICY UPDATE

Leveraging the GMM-based distributional safety critic, we propose a novel safety metric, $\text{SCVaR}_\alpha$, to guide safe exploration and improve policy optimization. At each iteration, the GMM parameters, $\Gamma^\pi(s, a) = \{(\mu_k, \sigma_k, \omega_k)\}_{k=1}^K$, are estimated, enabling the computation of the safety measure $\Lambda^\pi(s, a, \alpha)$ for a specified risk level $\alpha$:

$$\Lambda^\pi(s, a, \alpha) \doteq \text{SCVaR}_\alpha$$
$$= \sup_{k \in \{1, \dots, K\}} \text{CVaR}_\alpha^{(k)}$$
$$= \sup_{k \in \{1, \dots, K\}} \left( \mu_k + \sigma_k \frac{\phi\left(\Phi^{-1}(\alpha)\right)}{1 - \alpha} \right).$$

The policy is optimized under the constraint:

$$\Lambda^\pi(s, a, \alpha) \leq d, \quad \forall t,$$

where $d$ is a predefined safety threshold.

To achieve a balance between performance and safety, inspired by the SAC framework (Haarnoja et al., 2018), we optimize the policy $\pi_\theta$ by minimizing the Kullback-Leibler (KL) divergence between the current policy and a safety-adjusted target distribution:

$$\min_\pi D_{\text{KL}} \left( \pi(\cdot \mid s_t) \,\middle\|\, \frac{\exp\left(\frac{1}{\lambda}\left(Q_r^\pi(s_t, \cdot) - \kappa \Lambda^\pi(s_t, \cdot, \alpha)\right)\right)}{Z^\pi(s_t)} \right),$$

where $Z^\pi(s_t)$ is the partition function ensuring normalization, $\lambda > 0$ represents the temperature parameter controlling entropy, and $\kappa > 0$ is the safety weight regulating the trade-off between maximizing rewards and adhering to safety constraints.

The KL divergence can be equivalently expressed as:

$$D_{\text{KL}} \left( \pi_\theta(\cdot \mid s_t) \,\middle\|\, \exp\left(\frac{1}{\lambda} X_{\alpha, \kappa}^{\pi_\theta}(s_t, \cdot) - \log Z^{\pi_\theta}(s_t)\right) \right)$$
$$= \mathbb{E}_{(s_t, a_t) \sim \rho_{\pi_\theta}} \left[ -\log \frac{\pi_\theta(a_t \mid s_t)}{\exp\left(\frac{1}{\lambda} X_{\alpha, \kappa}^{\pi_\theta}(s_t, a_t) - \log Z^{\pi_\theta}(s_t)\right)} \right]$$
$$= \mathbb{E}_{(s_t, a_t) \sim \rho_{\pi_\theta}} \left[ \log \pi_\theta(a_t \mid s_t) - \frac{1}{\lambda} X_{\alpha, \kappa}^{\pi_\theta}(s_t, a_t) + \log Z^{\pi_\theta}(s_t) \right],$$

where
$$X^{\pi_\theta}_{\alpha,\kappa}(s,a) = Q^\pi_r(s,a) - \kappa\Lambda^\pi(s,a,\alpha),$$

and $Q^\pi_r(s,a)$ denotes the state-action value function.

As the partition function $Z^{\pi_\theta}(s_t)$ does not influence the gradient of $\theta$, it can be excluded from the optimization. This results in the actor loss function:

$$J_\pi(\theta) = \mathbb{E}_{(s_t,a_t)\sim\rho_{\pi_\theta}}\left[\lambda\log\pi_\theta(a_t\mid s_t) - X^{\pi_\theta}_{\alpha,\kappa}(s_t,a_t)\right].$$

To ensure the policy adheres to safety constraints, the safety weight $\kappa$ is adjusted dynamically by minimizing the following loss function:

$$J_s(\kappa) = \mathbb{E}_{(s_t,a_t)\sim\rho_{\pi_\theta}}\left[\kappa\left(d - \Lambda^\pi(s_t,a_t,\alpha)\right)\right],$$

where $d$ represents a predefined safety threshold.

This approach enables an adaptive trade-off between performance and safety by dynamically updating $\kappa$. The reward critic $Q^{\pi_\theta}_r$ and the entropy weight $\lambda$ are updated following the SAC method. Details on the loss functions $J_e(\lambda)$ for entropy adaptation and $J_r(\psi)$ for the reward critic can be found in Haarnoja et al. (2018).

### C.2 COMPLETE ALGORITHM: GMM-SSAC

See Alg. 3.

## D EXPERIMENT DETAILS

### D.1 TASK DESCRIPTION

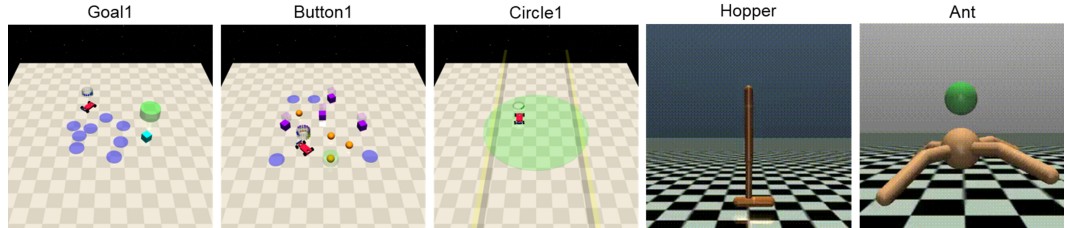

Figure 7: Illustration of five different tasks: Goal1, Button1, Circle1, Hopper, and Ant.

**Goal.** The agent's objective is to reach the goal buttons while avoiding static obstacles. Once the agent presses the correct button, a new goal button is randomly selected. The agent earns positive rewards for moving toward the goal and a bonus for successfully reaching it. Penalties are applied as costs for violating safety constraints, such as colliding with static obstacles or pressing the wrong button. The observation space includes the agent's ego states and sensory information about the obstacles and the goal, represented by pseudo LiDAR points. We use a Car robot in this environment and set the difficulty to level 1, naming it *CarGoal1*.

**Button.** This task is a more challenging version of Goal, featuring dynamic obstacles in addition to static ones. The dynamic obstacles move continuously along circular paths, requiring the agent to navigate to the goal while avoiding both static and dynamic obstacles. Compared to Circle and Goal tasks, Button demands greater inference capabilities as the agent must deduce the states of surrounding obstacles from raw sensory data. We use a Car robot and set the difficulty to level 1, naming it *CarButton1*.

**Circle.** In this task, the agent controls a robot to move clockwise along a circular path. Rewards increase as the agent's velocity rises and it stays closer to the circle's boundary. The safety zone is defined by two parallel plane boundaries intersecting the circle, and the agent incurs a penalty of 1 for leaving this zone. The observation space includes the car's ego states and sensory information

---

**Algorithm 3** GMM-SSAC: Gaussian Mixture Model-Based Supremum CVaR-Guided Safe Soft Actor-Critic

---

1: **Input:** Initial policy $\pi_\theta$, safety critic parameters $\phi$, reward critic parameters $\psi_1, \psi_2$, entropy weight $\lambda$, safety weight $\kappa$, risk level $\alpha$, safety threshold $d$, learning rates $\eta_\theta, \eta_\phi, \eta_\kappa$, target smoothing factor $\tau$, and replay buffer $\mathcal{D}$.

2: Initialize target networks for safety and reward critics with parameters $\phi', \psi_1', \psi_2'$.

3: **while** not converged **do**

4:     Sample action $a_t \sim \pi_\theta(\cdot \mid s_t)$, observe next state $s_{t+1}$, reward $r_t$, and cost $c_t$.

5:     Store transition $(s_t, a_t, r_t, c_t, s_{t+1})$ in replay buffer $\mathcal{D}$.

6:     **for** each gradient step **do**

7:         Sample a mini-batch of transitions $(s, a, r, c, s') \sim \mathcal{D}$.

        **Update Safety Critic:**

8:         Perform incremental Bellman update for safety critic:

$$\Psi_{\text{update}}(s, a) = \mathcal{R}(\Psi(s, a), \Psi_{\mathcal{B}}(s, a), \beta)$$
$$= \{x_1, \ldots, x_{M_1}, c(s, a) + \gamma x_1', \ldots, c(s, a) + \gamma x_{M_2}' \mid M_1 : M_2 = (1 - \beta) : \beta\}$$

9:         Fit GMM parameters $\Gamma_{\text{update}}^\pi(s, a) = \{(\mu_k^{\text{update}}, \sigma_k^{\text{update}}, \omega_k^{\text{update}})\}_{k=1}^K$ to $\Psi_{\text{update}}(s, a)$.

10:       Update safety critic network parameters $\phi$ by minimizing the MSE loss:

$$\mathcal{L}_{\text{safety}} = \sum_{k=1}^K \left[ (\mu_k - \mu_k^{\text{update}})^2 + (\sigma_k - \sigma_k^{\text{update}})^2 + (\omega_k - \omega_k^{\text{update}})^2 \right].$$

        **Update Reward Critic:**

11:       Minimize Bellman residuals for $Q_r^{\pi_\theta}$ using double-Q learning.

12:       **Compute SCVaR:**

$$\Lambda^\pi(s, a, \alpha) = \sup_{k \in \{1, \ldots, K\}} \left( \mu_k + \sigma_k \frac{\phi(\Phi^{-1}(\alpha))}{1 - \alpha} \right).$$

        **Update Policy:**

13:       Minimize actor loss:

$$J_\pi(\theta) = \mathbb{E}_{(s,a) \sim \mathcal{D}} \left[ \lambda \log \pi_\theta(a \mid s) - \left( Q_r^\pi(s, a) - \kappa \Lambda^\pi(s, a, \alpha) \right) \right].$$

        **Update Safety Weight:**

14:       Minimize safety loss:

$$J_s(\kappa) = \mathbb{E}_{(s,a) \sim \mathcal{D}} \left[ \kappa \big( d - \Lambda^\pi(s, a, \alpha) \big) \right].$$

15:       Perform gradient steps for $\theta, \kappa, \phi, \psi_1, \psi_2$.

        **Target Network Updates:**

16:       Update target networks:

$$\phi' \leftarrow \tau\phi + (1 - \tau)\phi', \quad \psi_1' \leftarrow \tau\psi_1 + (1 - \tau)\psi_1', \quad \psi_2' \leftarrow \tau\psi_2 + (1 - \tau)\psi_2'.$$

17:     **end for**

18: **end while**

19: **Output:** Optimized policy $\pi_\theta$.

---

about the boundary. We use a Car robot in this environment and set the difficulty to level 1, naming it *CarCircle1*.

**HopperVelocity.** This task requires the agent to control a hopper robot to move as quickly as possible while adhering to velocity constraints. Rewards are given for achieving high speeds, while penalties of 1 are applied if the velocity exceeds a predefined threshold, set to 50% of the hopper's maximum velocity determined after Proximal Policy Optimization (PPO) training for $10^7$ steps. This task emphasizes balancing speed optimization with safety constraints, naming it *Hopper*.

**AntVelocity.** Similar to HopperVelocity, this task involves controlling a quadruped ant robot under the same velocity constraints and reward structure. The velocity threshold is set to 50% of the ant's maximum velocity obtained after PPO training for $10^7$ steps. Due to the ant's higher degrees of freedom, this task presents additional challenges in balancing speed, stability, and adherence to safety constraints, naming it *Ant*.

## D.2 Implementation details & Hyper-parameter settings

**Compute Resources.** All experiments were run locally on a machine with an NVIDIA RTX 3090 GPU (24GB), 32GB RAM, and an Intel Core i7-12700KF CPU. The system ran Ubuntu 20.04 with Python 3.9, PyTorch 2.0, and CUDA 11.8. Each training run took 8–12 hours depending on the environment and GMM complexity, with total compute estimated at 4,000 GPU-hours. Early-stage experiments and failed runs are not included in this estimate.

**Baselines.** We use the official implementations from the respective codebases:

- WC-SAC(Yang et al., 2021): `https://github.com/AlgTUDelft/WCSAC`
- CAL(Wu et al., 2024): `https://github.com/ZifanWu/CAL`
- SAC & SAC-Lag: `https://github.com/PKU-Alignment/omnisafe` (a comprehensive framework for Safe RL algorithms (Ji et al., 2023))

We adopt the default hyper-parameter settings from the original implementations. Additionally, for WC-SAC, the risk hyperparameter $\alpha$ is set to 0.1. The safety threshold $d$ is configured as follows: 10 for CarGoal1 and CarButton1, and 25 for CarCircle1, Hopper, and Ant. These settings are consistent with the on-policy method CVPO (Liu et al., 2022) and the OmniSafe framework (Ji et al., 2023).

**GMM-SSAC.** The detailed settings for GMM-SSAC are summarized in Table 1.

Table 1: Hyper-parameter settings for GMM-SSAC.

| Parameter | Setting |
|---|---|
| Policy network sizes | [256, 256] |
| Q network sizes | [256, 256] |
| Network activation | ReLU |
| Discount factor $\gamma$ | 0.99 |
| Reward Critics learning rate | $1 \times 10^{-3}$ |
| Cost Critics learning rate | $1 \times 10^{-3}$ |
| Actor learning rate | $3 \times 10^{-4}$ |
| NN optimizer | Adam |
| Number of GMM components($K$) | 4 |
| Blending ratio ($\beta$) | 0.6 |
| Number of samples($M$) | 500 |

## D.3 Computational Efficiency Analysis

This section provides a detailed analysis of the computational overhead introduced by GMM-SSAC, complementing the runtime information. We examine both (i) the per-update theoretical complexity and (ii) the empirical wall-clock time across different GMM configurations.

**1. Theoretical Complexity.** GMM-SSAC introduces two additional operations beyond SAC-Lag: (i) fitting a GMM safety critic and (ii) performing a single-round EM update per critic step. Let $K$ denote the number of mixture components, $M$ the number of samples used in the EM update, and $H$ the forward/backward cost of the critic network. The per-update computational cost becomes:

$$\text{SAC-Lag: } 2H, \qquad \text{GMM-SSAC: } 2H + O(MK).$$

The $O(MK)$ term arises from the E-step (computing $\gamma_{mk}$ for $M$ samples and $K$ components) and the M-step (updating component-wise means, variances, and weights). Since $H$ typically dominates $MK$ in modern neural architectures, this overhead acts as a *constant-factor increase* and does not

grow with training steps. For moderate values of $K$, the added cost remains negligible relative to network forward/backward passes.

**2. Empirical Wall-Clock Measurements.** We benchmarked the full training time on a single RTX 3090 GPU for the CarGoal1 environment using 3M steps and a minibatch size of 256. The results are summarized below:

Table 2: Empirical wall-clock training time on a single RTX 3090 GPU (CarGoal1, 3M steps, batch size 256).

| Method | Time (hours) | Relative to SAC-Lag |
|---|---|---|
| SAC-Lag | 7.1 | 1.00× |
| GMM-SSAC (K=4) | 8.5 | 1.20× |
| GMM-SSAC (K=8) | 9.9 | 1.39× |

Across all environments, GMM-SSAC introduces a modest overhead of approximately **20%–40%** depending on $K$. Importantly, the EM update uses the *same minibatch* as the Bellman update and therefore does not require additional interactions with the environment, ensuring that *sample efficiency remains unchanged*.

Overall, the computational overhead introduced by GMM-SSAC is moderate, predictable, and well bounded. The method scales gracefully for small to medium numbers of mixture components, and the improvements in safety and tail-risk modeling are achieved with only a slight increase in training time.

### D.4 DERIVATION OF THE DISCOUNTED THRESHOLD

In Safe RL, the cost threshold $D$ ensures safety constraints during training. When using discounted costs, the total cost must account for the discount factor $\gamma$.

The discounted threshold $d$ adjusts the cost limit $D$ to reflect discounting. The total cost in an episode, discounted by $\gamma^t$, is:

$$C_{\text{Total}} = \sum_{t=1}^{T} \gamma^t \cdot C_t.$$

Assuming constant cost per time step $C_t = \frac{D}{T}$, we have:

$$C_{\text{Total}} = \frac{D}{T} \cdot \sum_{t=1}^{T} \gamma^t = \frac{D}{T} \cdot \frac{1 - \gamma^T}{1 - \gamma}.$$

Thus, the discounted threshold $d$ is:

$$d = \frac{D \cdot (1 - \gamma^T)}{(1 - \gamma) \cdot T}.$$

Here, $D$ is the cost limit, $\gamma$ is the discount factor, and $T$ is the maximum episode length (e.g., $T = 1000$).

The discounted threshold $d$ ensures the total cost stays within the original limit $D$, even with discounted future costs, and is essential for enforcing safety constraints in RL tasks.

# E   ABLATION STUDY

## E.1   SCVaR vs. CVaR

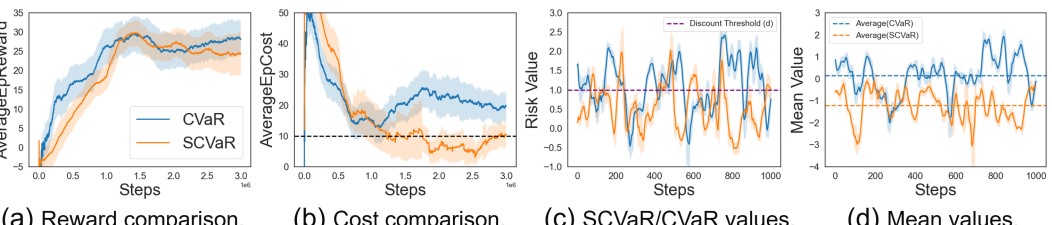

(a) Reward comparison.    (b) Cost comparison.    (c) SCVaR/CVaR values.    (d) Mean values.

Figure 8: Comparison of SCVaR and CVaR in terms of reward and cost performance, and GMM distribution.

We compare GMM-SSAC models trained in the CarGoal1 environment using SCVaR and Monte Carlo-based CVaR, as shown in Fig. 8.

To establish a fair baseline, CVaR is estimated empirically via Monte Carlo sampling. Given a learned GMM distribution

$$f_L(x) = \sum_{k=1}^{K} w_k \cdot \mathcal{N}(x \mid \mu_k, \sigma_k^2),$$

we draw $N = 5{,}000$ i.i.d. samples $\{x_j\}_{j=1}^{N}$ at each state-action pair. The empirical $\alpha$-level VaR is the $(1 - \alpha)$-quantile of the sampled values:

$$\widehat{\text{VaR}}_\alpha = \text{Quantile}_{1-\alpha}(\{x_j\}_{j=1}^{N}),$$

and the corresponding CVaR is the average of samples beyond this threshold:

$$\widehat{\text{CVaR}}_\alpha = \frac{1}{|\mathcal{I}|} \sum_{j \in \mathcal{I}} x_j, \quad \mathcal{I} = \{j \mid x_j \geq \widehat{\text{VaR}}_\alpha\}.$$

This approach avoids reliance on closed-form solutions and provides a flexible, data-driven estimate of tail risks. We adopt it as the practical CVaR baseline throughout our experiments.

Fig. 8(a) and Fig. 8(b) show that both models achieve similar rewards, but SCVaR achieves lower costs and consistently satisfies the safety threshold. To further understand this difference, both models were evaluated under identical random seeds and environment settings. Their estimated SCVaR (CVaR) values and the means of the GMM density functions are presented in Fig. 8(c) and Fig. 8(d). While the two methods yield similar tail risk estimates close to the discounted threshold in Fig. 8(c), SCVaR produces a smaller mean in Fig. 8(d), indicating that it enforces a stronger focus on the worst-case tail.

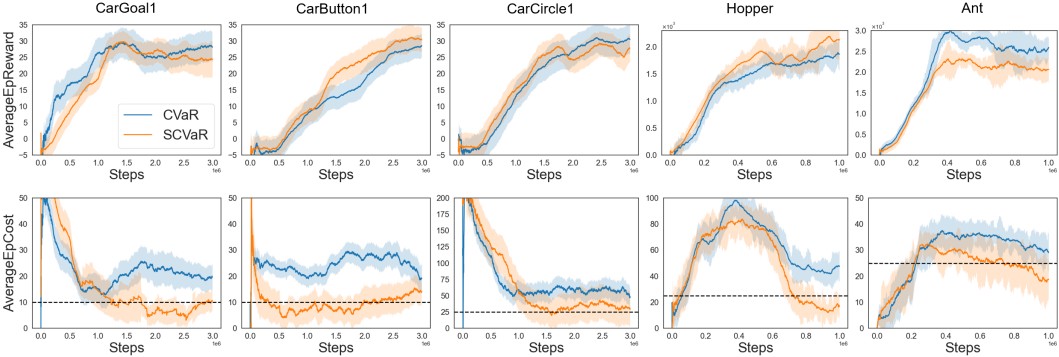

Figure 9: Additional experimental results for all environments.

Overall, these results demonstrate that SCVaR places greater emphasis on rare but extreme risks than Monte Carlo CVaR, leading to superior cost reduction even when such events occur with low probability. Consistent improvements across additional settings are reported in Fig. 9.

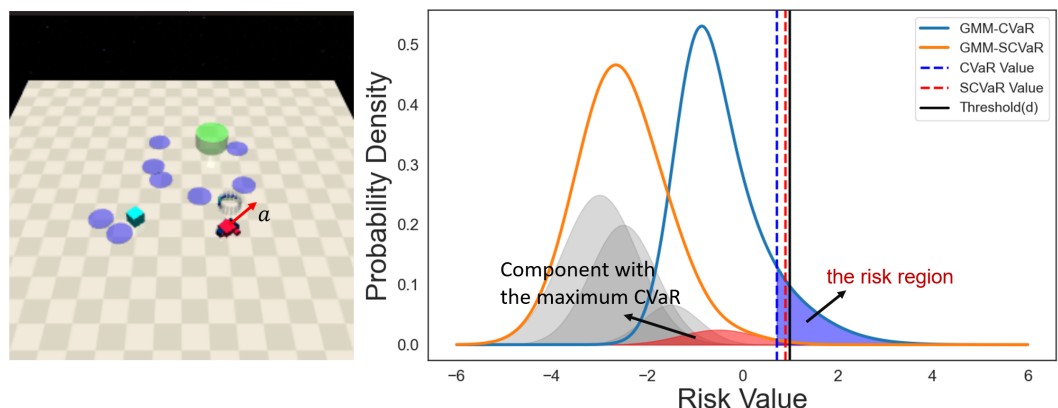

Figure 10: Visualization of the GMM cost distribution output by the Safety Critics in the SCVaR and CVaR models.

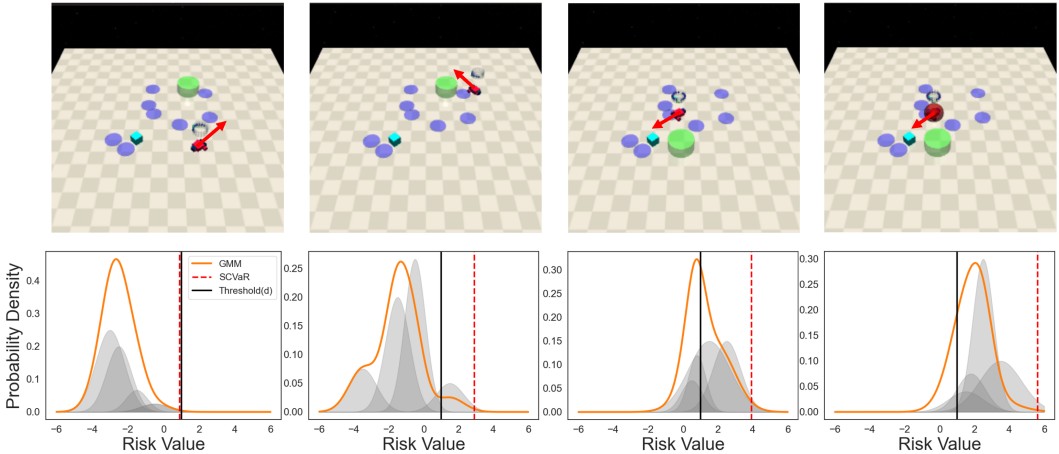

Figure 11: Cost distribution visualization for four scenarios.

### E.1.1 VISUALIZATION OF COST DISTRIBUTION

We input the same state-action pair into the Safety Critics of both models and visualize the GMM distributions they output, as shown in Fig. 10. Both models determine that the state-action pair is safe (SCVaR or CVaR $< d$). However, from the distribution curves, it is clear that SCVaR is much more conservative. It only considers the state-action pair safe when the high-risk region has almost no probability, and continuously adjusts its distribution during training to guide the policy network toward safer actions. In contrast, CVaR considers the state-action pair safe even when there remains a significant portion of high-risk areas, suggesting that this Safety Critic is not sufficiently "reliable."

Fig. 11 visualizes the cost distribution for four cases, representing safe, less safe, less dangerous, and dangerous scenarios. The SCVaR values accurately reflect risk levels, increasing with danger and exhibiting heavier upper tails. Note that the agent's policy is not optimal, as a fully converged policy would rarely encounter dangerous situations.

### E.2 GMM-BASED VARIANT IN BASELINES

We extend GMM-based safety critics to existing SafeRL baselines by replacing their original cost critics with GMM-based variants. Specifically, we implement GMM variants for SAC-Lag and CAL. Since SAC lacks a safety critic and WC-SAC with GMM + SCVaR already forms the basis of GMM-SSAC, no further variants are constructed for these algorithms.

We evaluate the modified baselines on two representative tasks—CarGoal1 from Safety-Gymnasium and Ant-Velocity from MuJoCo. As shown in Fig. 12, both SAC-Lag and CAL benefit significantly from the incorporation of GMM-based critics. In particular, the GMM variants yield higher reward performance while maintaining lower cumulative cost, indicating improved safety-reward trade-offs. This improvement is consistent across both tasks, suggesting that GMM critics can serve as a plug-in module to enhance the performance of diverse SafeRL frameworks.

These findings demonstrate that the use of GMM-based safety critics is not limited to our proposed method but can generalize to other constrained RL algorithms, offering a principled and effective alternative to conventional cost estimators.

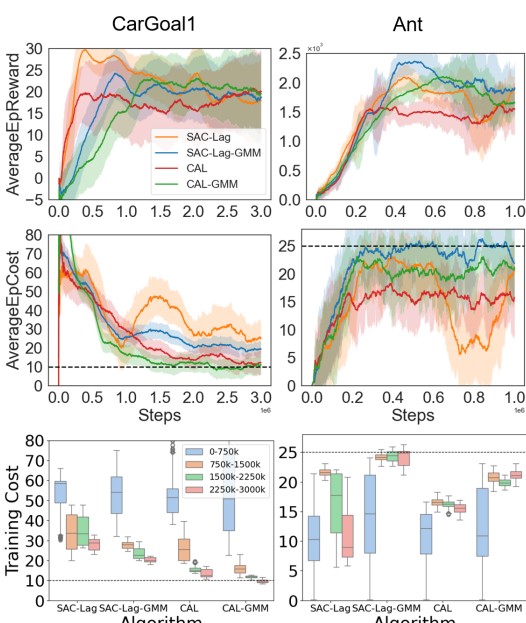

Figure 12: Performance comparison of SAC-Lag and CAL before and after substituting their cost critics with GMM-based variants.

### E.3 COMPONENT-LEVEL INTERPRETABILITY

We analyze whether GMM-based SCVaR components align with specific safety violations, as shown in Fig. 13. The CarButton1 environment was selected for its clear delineation of different safety risks, with three distinct violation types: **Gremlins** (contact-based penalties), **Wrong Buttons** (costs from incorrect button presses), and **Hazards** (proximity-based risks). This setup allows us to evaluate the relationship between GMM components and different violation types.

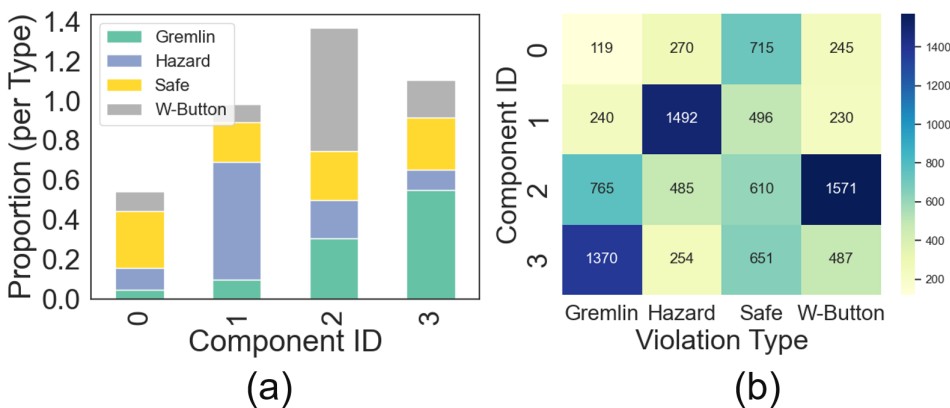

Figure 13: SCVaR component activation across safety violation types. (a) Normalized stacked bar chart of component dominance per violation type; (b) Heatmap of sample counts per component-violation pair. Components 3/2/1 correspond to Gremlins, Wrong Buttons, and Hazards respectively, while Component 0 serves as a fallback across mixed cases.

In total, we evaluate 10,000 $(s, a)$ samples, with 2,500 samples per category: Safe, Gremlins, Wrong Buttons, and Hazards. Fig. 13(a) shows the normalized proportion of each SCVaR component acti-

vated under these violation types, while Fig. 13(b) visualizes the sample counts with a heatmap. The results show clear alignment between components and violation types: Component 3 predominantly captures Gremlins, Component 2 specializes in Wrong Buttons with some overlap with Gremlins, Component 1 is mainly associated with Hazards, and Component 0 is more evenly distributed across all types, acting as a fallback in ambiguous cases. These findings highlight that GMM-based SCVaR naturally separates distinct violation types without explicit supervision, demonstrating its potential for uncovering structured risk semantics in safety-critical environments. Further visualization are provided in Fig. 14.

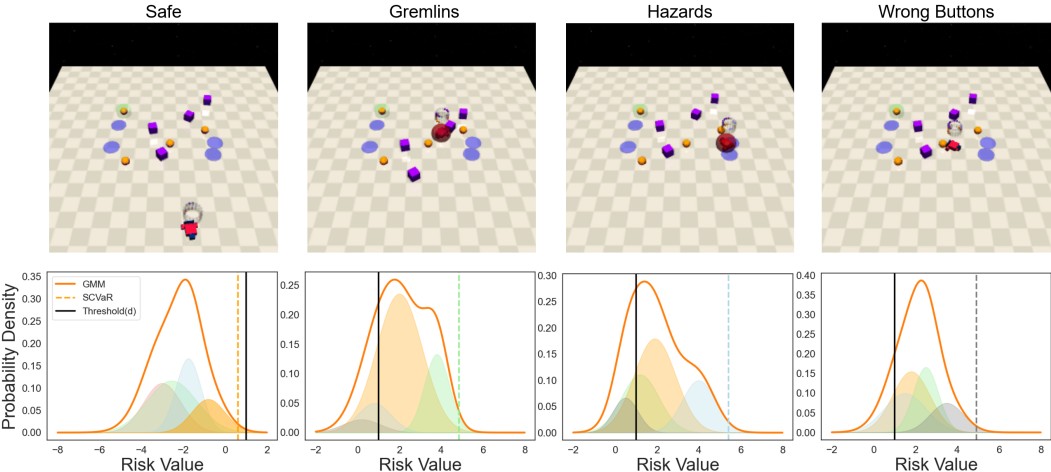

Figure 14: Cost distribution and GMM component visualization for four representative states in the CarButton1 environment. Each GMM component is assigned a consistent color across subplots (e.g., components 0–3 are colored orange, blue, gray, and green, respectively), and grouped according to their output order. The figure reveals that different types of safety-violating states are primarily associated with distinct GMM components, as indicated by color-coded contributions to SCVaR. This highlights the ability of GMM-based satety critics to disentangle different risk patterns via distinct components.

### E.4 POLICY-INDUCED DIFFERENCES IN RISK DISTRIBUTIONS

A natural question is whether the safety-cost distribution is truly multimodal or whether such structure may arise from function-approximation artifacts. Crucially, our method does not assume any particular form for the underlying distribution. The safety critic models the *risk-value distribution* $\mathcal{G}^\pi(s, a)$, which depends entirely on the trajectories induced by the policy $\pi$. Since different policies generate different occupancy measures, they necessarily induce different risk-value distributions—even including degenerate cases (e.g., a highly conservative policy may remain stationary, yielding a distribution concentrated near zero). A GMM critic is therefore well suited for this setting, as it provides a flexible nonparametric approximation capable of capturing unimodal, multimodal, skewed, or heavy-tailed structures without imposing any prior assumptions.

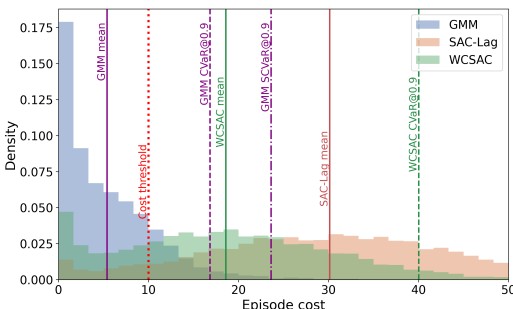

Figure 15: Empirical episode-cost distributions for three policies. GMM-SSAC substantially reduces tail mass and concentrates the distribution below the threshold.

To empirically validate that different policies indeed induce distinct risk distributions, we train three representative Safe RL algorithms on CarButton1: SAC-Lag (mean cost objective), WCSAC (Gaussian CVaR objective), and GMM-SSAC (SCVaR objective). After convergence, we fix the same

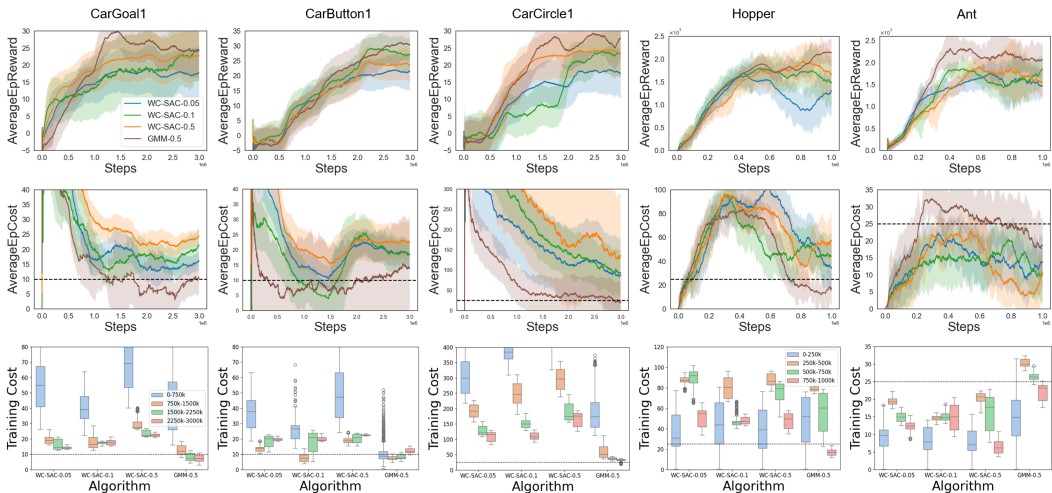

Figure 16: Performance of WC-SAC under different CVaR levels ($\alpha = 0.05, 0.1, 0.5$) compared with GMM-SSAC. Across all environments, none of the WC-SAC variants are able to reduce the average episodic cost below the safety threshold, even when using very small $\alpha$. In contrast, GMM-SSAC consistently maintains costs within the safe region, demonstrating that adjusting $\alpha$ cannot replicate the worst-component tail suppression achieved by SCVaR.

reference state–action pair $(s, a)$ and perform 10,000 Monte–Carlo rollouts for each policy to obtain an empirical estimate of $\mathcal{G}^\pi(s, a)$. The resulting histograms in Fig. 15 exhibit strikingly different shapes. SAC-Lag produces a wide distribution with a heavy upper tail, consistent with the fact that regulating only the expectation fails to constrain low-probability catastrophic outcomes. WCSAC reduces part of this tail but remains limited by its unimodal Gaussian assumption, leading to a residual probability mass well above the safety threshold. In contrast, GMM-SSAC produces a much more compact distribution whose empirical mean lies well below the safety threshold. Although the resulting CVaR and SCVaR values remain slightly above the threshold, the tail mass has been substantially pushed downward, bringing the upper tail much closer to the safe region. This represents a clear improvement over both SAC-Lag and WCSAC, whose unimodal Gaussian formulations leave a significantly larger portion of the distribution extending far beyond the threshold.

Overall, these observations confirm that distinct policies induce distinct risk-value distributions, and that different risk objectives shape these distributions in systematically different ways. More importantly, they demonstrate that combining a flexible GMM critic with the SCVaR objective provides robust tail-risk control and a more faithful representation of the underlying safety structure, irrespective of whether the true distribution is unimodal or multimodal.

### E.5 CAN A LOWER CVAR LEVEL REPLICATE THE BEHAVIOR OF SCVAR?

To investigate this, we conduct a risk-level ablation on WC-SAC with $\alpha \in \{0.05, 0.1, 0.5\}$ and compare the resulting behaviors with those of GMM-SSAC. As shown in Fig. 16, lowering $\alpha$ does make WC-SAC marginally more conservative; however, all WC-SAC variants still exhibit substantial safety violations across environments, and none of them succeed in driving the average episodic cost below the task-specific threshold. In contrast, GMM-SSAC consistently maintains costs below (or tightly around) the safety limit, demonstrating a qualitatively stronger form of tail-risk control.

This consistent gap arises from a fundamental modeling limitation of single-Gaussian critics. Reducing $\alpha$ merely probes deeper into the tail of a collapsed Gaussian approximation, whose shape is dictated primarily by its global mean and variance. Because rare but catastrophic trajectories contribute negligibly to these parameters, the resulting Gaussian CVaR systematically underestimates the true upper tail and fails to impose sufficient penalty on high-cost modes. Consequently, even with $\alpha = 0.05$, WC-SAC remains unable to suppress the long tail of the cost distribution enough to satisfy the safety constraint.

In contrast, SCVaR considers the worst tail among all mixture components, preserving catastrophic modes rather than averaging them away. This mixture-aware formulation allows GMM-SSAC to detect high-cost components even when their probability mass is extremely small, and to reshape the entire risk-value distribution by aggressively pushing down its tail. Empirically, this leads to a consistent reduction of average costs below the constraint threshold—an effect that cannot be recovered by tuning $\alpha$ within a unimodal critic.

### E.6 CHOICE AND ADAPTIVITY OF THE RISK LEVEL $\alpha$.

We provide additional analysis on how to choose the SCVaR risk-level parameter $\alpha$, whether extensive tuning is required, and how an adaptive version behaves in comparison with fixed choices. The empirical results in Fig. 17 show that $\alpha$ has clear physical meaning, produces smooth safety–performance trade-offs, and requires only coarse selection in practice.

In realistic usage, larger values (e.g. 0.9) emphasize shallower within-mode tails and favor higher returns but incur higher costs, whereas smaller values (e.g. 0.1) probe deeper tails and enforce strong conservativeness. As seen in the reward and cost curves, this effect is monotone and predictable: $\alpha = 0.1$ yields the lowest episodic costs, while $\alpha = 0.9$ achieves higher rewards but substantially more constraint violations.

We also implemented a preliminary *adaptive-$\alpha(s)$ variant* in order to examine whether dynamically adjusting the risk level can further improve SCVaR-based control. In this variant, the policy network is augmented with an additional output head that predicts a state-conditioned risk level $\alpha(s) \in (0,1)$ alongside the action distribution. During training, $\alpha(s)$ is treated as a differentiable policy parameter: for each sampled transition $(s,a)$, we compute $\mathrm{SCVaR}_{\alpha(s)}(\mathcal{G}^\pi(s,a))$ and backpropagate its gradient through both the action and the $\alpha(s)$ branch of the actor. This design effectively allows the agent to modulate its risk sensitivity on a per-state basis, increasing conservativeness in hazardous regions and relaxing it elsewhere.

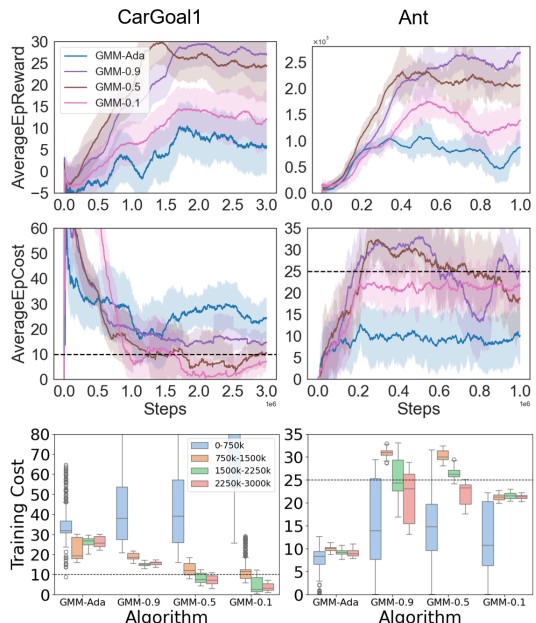

Figure 17: Comparison of fixed-$\alpha$ and adaptive-$\alpha(s)$ variants on CarGoal1 and Ant, showing reward curves, cost curves, and training-phase cost distributions. Fixed values ($\alpha = 0.1, 0.5, 0.9$) produce smooth, monotone safety–performance trade-offs, whereas the adaptive variant displays higher variance and less stable risk control.

While this formulation is flexible, Fig. 17 shows that the adaptive-$\alpha(s)$ variant exhibits noticeably higher variance in both reward and cost, as well as weaker asymptotic safety compared to fixed $\alpha$. We attribute this to two factors: (i) the joint optimization of $(a, \alpha(s))$ increases gradient noise and destabilizes early training, and (ii) SCVaR is non-convex with respect to $\alpha$, making the $\alpha(s)$ branch more difficult to optimize reliably. Consequently, although adaptive-$\alpha$ is conceptually appealing and represents a promising direction for future work, the fixed-$\alpha$ formulation is empirically more stable, easier to train, and provides stronger safety–performance trade-offs in our current framework.

### E.7 ADDITIONAL BASELINES: EXTENSION OF SCVAR TO PPO-BASED SAFE RL METHODS

The baselines considered in the main text focus on off-policy actor–critic methods built upon the SAC framework. This design choice ensures that all compared algorithms share a consistent optimization pipeline (soft policy iteration, entropy-regularized objectives, and off-policy training) and differ only in their risk modeling and constraint-handling mechanisms. To further examine the gen-

erality of our approach beyond the SAC family, we additionally extend SCVaR to the on-policy PPO regime and compare it against several widely studied PPO-based safe RL algorithms.

**Overview of Compared PPO-Based Safe RL Methods.**    We briefly summarize the baselines included in this ablation:

- **CPPO** (Ying et al., 2022) imposes trust-region constraints on policy updates to guarantee satisfaction of *expected* cost constraints.
- **CVaR-CPO** (Zhang et al., 2024a) extends CPO by incorporating a CVaR-based constraint, using quantile-based estimation of CVaR and a trust-region update scheme.
- **SDAC** (Kim et al., 2023) utilizes a mean–variance surrogate and dual updates to control risk sensitivity.
- **SDPO** (Zhang & Weng, 2021) extends IPO to the distributional RL setting by learning full return distributions via quantile regression, enabling more expressive constraints such as CVaR.
- **SRCPO** (Kim et al., 2024) optimizes a CVaR-like surrogate via primal–dual updates and proves convergence to a risk-constrained optimal policy.

These approaches collectively provide a representative comparison set within the PPO framework, covering expected constraints, CVaR-based constraints, variance-based risk, and distributional constraints.

**Our PPO Variant: GMM-SPPO.**    Because SCVaR is defined purely at the level of the value distribution, it is not tied to SAC and can be integrated into any actor–critic framework that maintains a safety critic. We therefore construct a PPO-based variant:

$$\text{GMM-SPPO} := \text{PPO} + \text{GMM safety critic} + \text{SCVaR}_\alpha \text{ constraint.}$$

The safety critic models the full return distribution using a Gaussian mixture with $K$ components, and the policy update uses the SC-VaR penalty or constraint in place of traditional expected-cost terms. No changes to PPO's clipped objective or trust-region structure are required.

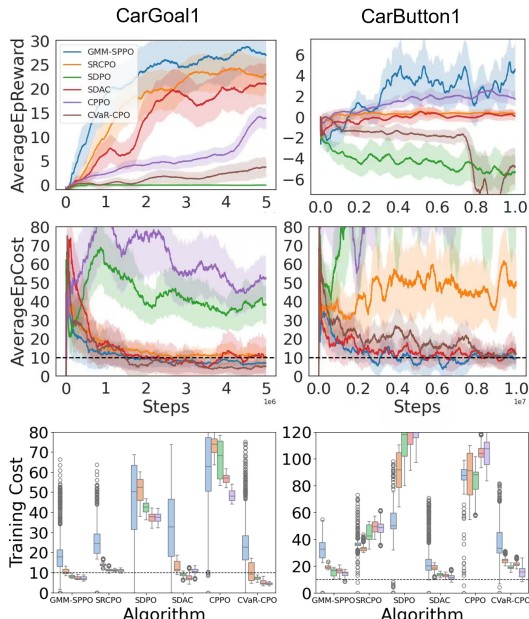

**Experimental Setup.**    Following prior work, particularly SRCPO whose original experiments and public code support only two task families (Goal and Button) instantiated in four environments (PointGoal1, PointButton1, CarGoal1, and CarButton1), we evaluate all PPO-based baselines on the two car-based environments CarGoal1 and CarButton1. All methods are trained for an equal number of steps with identical random seeds, evaluation protocols, and safety thresholds to ensure comparability.

Fig. 18 presents the corresponding reward–cost trajectories. Across both environments, GMM-SPPO consistently achieves higher asymptotic rewards while maintaining strictly lower accumulated costs throughout training. Its safety performance remains high and stable, in contrast to several PPO-based baselines whose behaviors vary substantially over time. These trends suggest that the combination of a

Figure 18:    Performance of GMM-SPPO compared with PPO-based safe RL baselines on Car-Goal1 and CarButton1.    GMM-SPPO attains higher rewards and lower costs, while several baselines exhibit constraint violations or degraded reward performance.

mixture-based safety critic and the SCVaR objective yields more robust tail-risk suppression and more reliable overall performance.

The PPO-based baselines display distinct safety and reward characteristics across the two tasks. CPPO and SDPO are generally unable to satisfy the safety constraints, exhibiting persistent violations throughout training. CVaR-CPO succeeds in keeping the average cost below the threshold but suffers from markedly reduced reward performance. SDAC and SRCPO perform comparably on CarGoal1, yet their behaviors diverge on CarButton1, where SRCPO fails to maintain the required safety level and accumulates costs above the allowed threshold. These observations highlight the sensitivity of PPO-based safe RL methods to environment dynamics and underscore the importance of accurately modeling the tail behavior of the safety-return distribution.

Overall, this ablation study demonstrates that the proposed SCVaR mechanism is not limited to off-policy SAC backbones but also provides substantial performance benefits within the PPO family. The mixture-based critic is crucial for capturing rare but catastrophic modes of the safety-return distribution, a capability that unimodal critics and CVaR-only constraints inherently lack.

## F   RELATED WORKS

### F.1   SAFE REINFORCEMENT LEARNING

Recent advances in Safe RL have introduced a variety of methods to ensure safety during RL training. Early Safe RL approaches were heavily influenced by control theory. Lyapunov functions are widely used in control theory to guarantee safety by constraining the action space during exploration (Chow et al., 2019; Huh & Yang, 2020; Jeddi et al., 2021). However, defining suitable Lyapunov functions often requires a system model, which may not be readily available in general RL scenarios. In contrast, Lagrangian-based approaches, particularly primal-dual optimization (Paternain et al., 2022), have gained significant attention due to their flexibility and broad applicability. These methods have been shown to achieve a zero duality gap under certain conditions, providing theoretical guarantees for constraint satisfaction (Paternain et al., 2019). Among these approaches, risk-constrained primal-dual methods (Chow et al., 2018) focus on developing efficient reinforcement learning algorithms for risk-constrained MDPs, where risk is typically represented through chance constraints or constraints on the CVaR of the cumulative cost. Additionally, reward-constrained methods (Tessler et al., 2018) utilize alternative penalty signals to guide the policy towards satisfying safety constraints. Alternatively, robust MDP methods (Iyengar, 2005; Wang & Zou, 2021) aim to learn policies that perform well under worst-case transition dynamics, but they often lead to overly conservative strategies and require specifying uncertainty sets. Furthermore, the Natural Policy Gradient Primal-Dual (NPG-PD) method (Ding et al., 2020) is the first to establish non-asymptotic convergence guarantees.

Similarly, our work builds upon primal-dual optimization methods. Among the most relevant recent developments, two works are particularly aligned with our framework. WCSAC (Yang et al., 2021) estimates the risk distribution using a unimodal Gaussian and extends the SAC-Lag method by incorporating a variance estimator to enhance risk control. CAL (Wu et al., 2024), on the other hand, addresses cost underestimation by employing an upper confidence bound (UCB) for the cost value, thereby improving risk management during policy optimization. However, these methods rely on Gaussian distributions to approximate risk distributions, overlooking the inherent limitations in their expressiveness.

**Distributionally Robust CMDPs**. A complementary line of work introduces distributional robustness on top of safety constraints. Building on robust and distributionally robust MDPs, which optimize the worst-case return over an ambiguity set of transition kernels (Xu & Mannor, 2010; Iyengar, 2005; Wiesemann et al., 2013), Russel et al. (2020) formulate robust constrained MDPs that merge CMDP and robust MDP frameworks to obtain policies that satisfy constraints even under transition-model misspecification. More recently, Zhang et al. (2024b) study distributionally robust constrained reinforcement learning under strong duality and develop algorithms with end-to-end guarantees for a class of environmental uncertainty sets.

In contrast, our approach does not posit an ambiguity set over transition dynamics and is therefore not a distributionally robust CMDP in the classical sense. DR-CMDP methods explicitly construct uncertainty sets over dynamics or costs (e.g., Wasserstein or f-divergence balls) and optimize worst-case return or constraint values under those sets, targeting robustness to model misspecification or adversarial perturbations. While SCVaR exhibits a "robust" flavor by taking the worst CVaR across

mixture components, this robustness is fundamentally mode-wise: we fit a parametric cost-return distribution $(\mathcal{G}(s,a))$ from empirical samples and exploit its mixture structure to identify the most hazardous latent mode. SCVaR thus enhances sensitivity to multi-modal tail risks without assuming environment perturbations or performing minimax optimization. Because safe RL benchmarks such as Safety Gymnasium operate under a fixed and stationary environment, where the key challenge is accurate tail-risk modeling rather than uncertainty in transition dynamics, SCVaR targets a different failure mode than DR-CMDP approaches. As a result, DR-CMDP algorithms do not constitute directly comparable baselines for our problem setting, and our empirical evaluation focuses on CMDP-based methods (e.g., SAC-Lagrangian, CVaR-based critics, actor-penalty approaches) that share the same modeling assumptions and evaluation protocol.

**Risk-Sensitive and Distributional Safe RL Beyond CVaR**. Beyond classical Gaussian-based cost critics, several recent works explore alternative ways to model safety risk. Off-policy TRC (Kim & Oh, 2022) stabilizes CVaR-constrained learning under off-policy data using trust-region surrogate bounds, but it still relies on a unimodal Gaussian assumption for cumulative costs, limiting its ability to capture multi-modal tail risks. CPPO (Ying et al., 2022) integrates CVaR regularization into PPO but similarly treats tail risk as arising from a single underlying distribution, preventing differentiation between distinct hazard modes. CVaR-RF-UCRL (Ni et al., 2024) takes a distributionally robust perspective by optimizing worst-case CVaR over an ambiguity set of transition kernels, addressing model uncertainty rather than the structure of the cost-return distribution itself, and therefore targets a fundamentally different problem setting. Alternative coherent risk measures such as EVaR and its ERM variants (Ni & Lai, 2022; Su et al., 2024) provide exponential-moment-based upper bounds for tail events, but their exponential-tail bias often leads to overly conservative behavior and lacks interpretability with respect to heterogeneous risk sources.

In contrast, our method neither assumes unimodal tails nor introduces transition-level ambiguity sets. Instead, it explicitly models the multi-modal structure of the cost-return distribution via a GMM and uses SCVaR to target the worst latent risk mode, enabling fine-grained and structurally aware control of safety-critical tail behavior.

## F.2 GMMs in RL

GMMs have proven to be powerful tools for modeling complex and multimodal distributions in reinforcement learning. Early work demonstrated that GMMs can approximate joint densities over continuous state–action–value triplets, offering a flexible alternative to kernel-based or parametric value approximators (Agostini & Celaya, 2010). In policy optimization, mixture-based policies parameterized by GMMs enable agents to switch among different behavioral modes and capture richer stochasticity and robustness (Haarnoja et al., 1861; Baram et al., 2021; Kim et al., 2022). More expressive policy classes such as diffusion-based RL also benefit from GMMs, where a mixture fitted to sampled actions provides a tractable surrogate for entropy estimation (Wang et al., 2024).

In value function approximation, GMM-based Q-functions integrate mixture likelihoods directly into the Bellman residual and have been shown to improve representational fidelity and numerical stability (Vu & Slavakis, 2024). Beyond parametric mixture critics, distributional RL has introduced alternative parameterizations, including categorical critics in (Bellemare et al., 2017), quantile regression critics (Dabney et al., 2018), and moment-parameterized critics (Cho et al., 2024). While expressive, these methods typically represent the return distribution as a single aggregated object rather than as decomposable modes, limiting their ability to support mode-wise tail-risk analysis.

A representative mixture-based distributional approach is GMAC (Nam et al., 2021), which also employs a GMM critic but optimizes it through energy distance (Cramér distance) minimization and a sample-replacement operator $SR(\lambda)$. GMAC shows that combining energy-distance gradients with $SR(\lambda)$ produces stable mixture-space Bellman updates, allowing the GMM critic to converge toward a fixed point and improving reward-return estimation. However, GMAC is designed for reward distribution modeling and does not aim to preserve mixture components for risk interpretation. In contrast, our Bellman–EM refinement maintains mode separability under off-policy sampling, which is crucial for analyzing heterogeneous safety hazards. Moreover, SCVaR evaluates the worst per-component CVaR, rather than approximating the full return distribution as in GMAC.

In summary, although our work builds on the expressive distribution modeling capabilities of GMMs, it extends their use to the risk-sensitive safety-value setting and introduces SCVaR and Bellman–EM refinement to enable principled mode-wise tail-risk control.

## G  THE USE OF LARGE LANGUAGE MODELS (LLMS)

We acknowledge the use of large language models (LLMs) in preparing this paper. LLMs were employed exclusively as an assistive tool for language refinement, including improving clarity, grammar, and readability of the manuscript. They were not used for research ideation, algorithm design, experimental implementation, or result generation. All technical contributions, theoretical analyses, and experimental results are solely the work of the authors.

The authors take full responsibility for the correctness and integrity of the paper's content.

