# OpenReview forum: "Focusing on the Riskiest: Gaussian Mixture Models for Safe Reinforcement Learning"
_ICLR.cc/2026/Conference — Submitted to ICLR 2026_

### Official Review · Reviewer_MMLH · 2025-10-20

**Soundness:** 3
**Presentation:** 3
**Contribution:** 3
**Rating:** 4
**Confidence:** 3

**Summary:**

This work mainly considers safe RL with safety constraints. Specially, CVaR may fail to capture complex, multimodal, or heavy-tailed risks, thus this work proposes the Supremum Conditional Value‑at‑Risk (SCVaR) for capturing worst‑case tail across all components of a Gaussian mixture. Consequently, combining with an EM‑based method to update the GMM parameters, the proposed GMM‑SSAC(Gaussian Mixture Model‑Based Supremum CVaR‑Guided Safe Soft Actor‑Critic) can estimate reliable SCVaR. Extensive theoretical and experimental results show that GMM‑SSAC is better than previous CVaR-based RL methods.

**Strengths:**

- I really like the idea of introducing new metrics in safe RL, thus I think SCVaR is a clear contribution if authors can introduce its advantages in safe RL more clearly (see weakness).

- SCVaR has some obvious insights like it can be easily computed in GMM, which is an expressive distributional framework.

- Lots of theoretical analyses clearly state the properties of SCVaR.

- The writing is great and easy to read.

**Weaknesses:**

- My major concern is, what is the main advantages of SCVaR compared with CVaR? In my understanding, CVaR considers the tail of the distribution and SCVaR considers the maximum CVaR of each component. What are the benefits of ignoring the tail distribution of other components? Providing some theoretical or experimental insight will make this work more solid. Also, a natural question is, can SCVaR be extended to any distributions that can not be represented by GMM? As CVaR is well defined in all distributions, the application of SCVaR will be limited if it can only be considered on GMM (of course CVaR of complex distribution can not be directly computed but still can be estimated).

- Assume that the ground truth distribution is GMM, there are always estimated gap between our estimated GMM and the ground truth distribution. Under this situation, what about the relationship of the estimated SCVaR and the ground truth SCVaR?

- Assume that the ground truth distribution is **not** GMM, of course we can utilize a GMM to estimate this distribution and calculated our estimated SCVaR, then what is the meaning of this estimated SCVaR?

- In experiments, I think a natural ablation study is that utilizing GMM to estimate the distribution and use CVaR of the estimated GMM to measure the risk, which might be a good comparison of SCVaR and CVaR.

- There are several works on safe RL with different risk measures need to be discussed, like CVaR [1-3] and EVaR [4-5].

Overall, I think this work is currently boardline, I'd like to actively join in the following discussion and adjust my score if the authors can address my concerns.

Ref:

[1] Towards safe reinforcement learning via constraining conditional value-at-risk

[2] Efficient off-policy safe reinforcement learning using trust region conditional value at risk

[3] Risk-sensitive reward-free reinforcement learning with cvar

[4] Risk-sensitive reinforcement learning via Entropic-VaR optimization

[5] Evar optimization in mdps with total reward criterion

**Questions:**

See weaknesses above

---

> ### Author Response · Authors · 2025-11-28
> **(1/4)**
>
> We appreciate the reviewer’s question and provide a detailed explanation below.
>
> ---
>
> **W1. “What are the main advantages of SCVaR compared with CVaR?”**
>
> **Reviewer’s concern.**
> * *What is the main advantage of SCVaR over CVaR?*
> * *Does SCVaR “ignore” other mixture components? Why is this justified?*
> * *Is SCVaR limited to GMMs? Can it be extended to other distributional models?*
>
>
> **Re_W1.** We thank the reviewer for the thoughtful questions and respond to each concern in detail.
>
> **1. Main advantage of SCVaR over standard CVaR.**
>
> A fundamental limitation of CVaR under mixture distributions is that mixture CVaR is necessarily a **convex aggregation** of component-wise CVaRs(Appendix A.1):
> $CVaR _\alpha(Z) = \sum _{k=1}^K {PostWeight}_i \mathrm{CVaR} _\alpha(Z_k).$
>
> Therefore, CVaR is dominated by high-probability components and inevitably **underestimates rare but catastrophic modes**, even for very small $\alpha$. SCVaR directly corrects this structural weakness by computing $\mathrm{SCVaR} _\alpha = \max _k\mathrm{CVaR} _\alpha(Z_k),$ which preserves the worst-mode tail and provides the **tightest coherent upper bound** of mixture CVaR. This guarantees sensitivity to catastrophic modes, which is crucial in Safe RL because a policy must remain safe under all plausible modes rather than only the most probable one.
>
> **2. SCVaR does *not* ignore other components.**
> All mixture components are fully used in (i) **learning** and (ii) **risk evaluation**.
> During learning, each component’s parameters $(\mu_k,\sigma_k,\omega_k)$ are updated using full EM responsibilities $\gamma_{mk}=\frac{\omega_k\mathcal N(z_m|\mu_k,\sigma_k)}{\sum_j\omega_j\mathcal N(z_m|\mu_j,\sigma_j)},$ ensuring no component is discarded or neglected. During risk evaluation, SCVaR computes $\mathrm{CVaR}_\alpha(Z_k)$ for **all** components and then applies a mode-wise dominance test. The max operator ensures that catastrophic components are preserved rather than masked by convex averaging, which is a known limitation of mixture CVaR.
>
> **3. Why this is theoretically justified.**
> In coherent risk-measure theory, SCVaR equals the **least conservative coherent majorant** of mixture CVaR. It is the smallest coherent measure that (i) upper-bounds mixture CVaR, and (ii) is robust to mixture-weight misspecification, since mixture CVaR’s convex structure systematically suppresses rare high-risk modes. SCVaR retains coherence (monotonicity, translation invariance, positive homogeneity, subadditivity), making it well suited for primal–dual CMDP optimization, unlike arbitrary conservative heuristics.
>
> **4. Supporting experimental evidence.**
> To directly address this concern, we added comparisons with WC-SAC under multiple CVaR levels. As shown in Appendix Fig.16, reducing $\alpha$ indeed makes WC-SAC slightly more conservative, but all variants continue to incur substantial safety violations across environments, and none are able to bring the average episodic cost below the threshold. In contrast, our GMM-SSAC consistently keeps costs below (or tightly around) the safety limit. This evidence confirms that merely lowering the CVaR confidence level cannot recover the level of tail-risk control achieved by our SCVaR-based approach.
>
> **5. SCVaR is *not* limited to GMMs.**
> (See also our response to reviewer S7ES.)
> The definition of SCVaR requires only the ability to compute or approximate **component-conditional tail risks**:
> $
> \mathrm{CVaR} _\alpha^{(k)}
> =\mathbb{E}[Z\mid Z\ge \mathrm{VaR} _\alpha^{(k)},\mathcal M _k].
> $
> Thus SCVaR is fully **model-agnostic**. It can be applied to:
>
> * **Normalizing flows:** via latent clustering or lightweight posterior GMMs.
> * **Diffusion models:** via posterior clustering or multi-head CVaR predictors.
> * **Autoregressive critics:** via multi-head mode-specific tail estimators.
>
> These alternatives, however, require numerical tail integration, expensive sampling, or unstable clustering, whereas GMMs offer (i) closed-form VaR/CVaR, (ii) a stable EM projection, (iii) tractable theoretical analysis (e.g., contraction), and (iv) efficient training.
>
> In short, SCVaR incorporates all mixture components and yields the minimal coherent upper bound that preserves worst-case tail behavior, while still being applicable to any model that can provide component-specific tail estimates.

---

> ### Author Response · Authors · 2025-11-28
> **(2/4)**
>
> **W2. “Relationship between estimated SCVaR and true SCVaR if ground truth is a GMM.”**
>
> **Reviewer’s concern.**
> If the ground-truth cost distribution is itself a GMM, what is the approximation gap?
>
> **Re_W2.** Let the ground-truth safety-cost distribution at $(s,a)$ be a Gaussian mixture $\mathcal{G}^\*(s,a)={(\mu_j^\*,\sigma_j^\*,\omega_j^\*)} _{j=1}^{K^\*},$
> and let $\mathcal{G}^{\pi}(s,a)$ denote the learned mixture generated by our Bellman–EM critic update.
>
> **1. Bellman–EM contraction ensures the learned GMM converges toward the ground-truth GMM**
>
> Although the model is trained on finite data and hence $\mathcal{G}_{\pi}\neq \mathcal{G}^{*}$, the Bellman–EM update is contractive:
>
> * The Bellman operator $T^\pi$ is a $\gamma$-contraction in $W_1$;
> * The EM projection $P$ is **non-expansive** in $W_1$.
>
> Therefore the composite operator satisfies
> $
> W _1\big( PT^\pi(\mathcal{G}^{\pi}), PT^\pi(\mathcal{G}^{\*})\big)
> \le
> \gamma W _1(\mathcal{G}^{\pi},\mathcal{G}^{\*}),
> \qquad \gamma\in(0,1).
> $
>
> Thus the parameter error between $\mathcal{G}_{\pi}$ and $\mathcal{G}^{*}$ decays **geometrically**, and the learned mixture converges to the true mixture whenever the model is expressive enough.
>
> **2. SCVaR is Lipschitz in the component parameters of the dominant tail mode**
>
> For any mixture $ \mathcal{G}={(\mu_k,\sigma_k,\omega_k)}$,
> $SCVaR_\alpha(\mathcal G)=\max_k(\mu_k+c_\alpha\sigma_k),\quad
> c_\alpha=\frac{\varphi(\Phi^{-1}(1-\alpha))}{1-\alpha}.$
>
> Two important facts follow:
>
> **a. SCVaR depends only on the dominant tail component**
>
> Let $j^\star = \arg\max_j (\mu_j^* + c_\alpha\sigma_j^*).$
> SCVaR ignores mixture weights and all non-dominant components.
>
> **b. Gaussian CVaR is Lipschitz in ((\mu_k,\sigma_k))**
>
> $
> \big|(\hat\mu-\mu^\*) + c_\alpha(\hat\sigma-\sigma^\*)\big|
> \le
> L_\alpha |(\hat\mu,\hat\sigma)-(\mu^\*,\sigma^\*)|.
> $
>
> **c. Max operator is non-expansive**
>
> $
> \big|\max_k a_k - \max_k b_k\big| \le \max_k |a_k - b_k|.
> $
>
> Therefore, combining (a)–(c), we obtain the stable SCVaR bound:
> $|\\mathrm{SCVaR} _\alpha(\mathcal G^\pi) - \mathrm{SCVaR} _\alpha(\mathcal G^\star)\|
> \le L _\alpha\ W _1(\mathcal G^\pi,\mathcal G^\star).$
>
>
> for a finite constant $L_\alpha$.
>
> As the Bellman–EM contraction drives $W_1(\mathcal{G}_{\pi},\mathcal{G}^{*})\to 0$,
> **the estimated SCVaR converges to the true SCVaR**.
>
> **3. Stability holds even when the number of mixture components is misspecified $(K \neq K^*)$**
>
> This is because SCVaR depends only on the **single** dominant tail mode.
>
> **Case 1: Learned GMM has more components $(K > K^*)$**
>
> EM typically **splits** a true component into several nearby subcomponents.
> Taking the maximum CVaR across components automatically preserves the true worst tail:
> $
> \max(\text{split modes}) = \text{true worst mode}.
> $
> No underestimation is possible.
>
> **Case 2: Learned GMM has fewer components $(K < K^*)$**
>
> EM must **merge** several true components into one broader Gaussian.
> This merged component generally has **larger variance**, therefore larger $\mu + c_\alpha\sigma.$
> Hence the resulting SCVaR is **conservative (safe)** but never optimistic.
>
> Thus, Model mismatch influences only critic granularity and may introduce mild conservatism,
> but **cannot lead to underestimation of catastrophic tail risk**.
>
> In summary:
>
> 1. The Bellman–EM operator guarantees that the learned mixture $\mathcal{G}_\pi$ converges toward $\mathcal{G}^*$.
> 2. SCVaR is a maximization of a linear tail statistic and is therefore Lipschitz and non-expansive.
> 3. The SCVaR error is upper-bounded by $O(W_1(\mathcal{G}_\pi,\mathcal{G}^*))$, which shrinks geometrically.
> 4. Even when $K\neq K^*$, SCVaR remains robust by preserving or overestimating the dominant tail mode.
>
> **Therefore, SCVaR is a stable, consistent, and non-optimistic estimator of the true tail risk when the ground truth is a GMM.**

---

> ### Author Response · Authors · 2025-11-28
> **(3/4)**
>
> **W3. “What if the true distribution is NOT a GMM?”**
>
> **Reviewer’s concern.**
> If the true distribution is not a GMM, what is the meaning of SCVaR of an approximated GMM?
>
> **Re_W3.** This is a natural question about representation error. Even if the ground-truth distribution $\mathcal{D}^\*$ is not a Gaussian mixture, the SCVaR computed on an approximating mixture $\widehat{\mathcal{G}}$ is still a well-defined and meaningful quantity. It can be interpreted from these complementary and rigorous perspectives:
>
> **1. $SCVaR(\widehat{\mathcal{G}})$ is a tail-risk functional applied to the best tractable approximation of $\mathcal{D}^*$**
>
> When $\mathcal{D}^*$ is arbitrary, a GMM serves as a **universal density approximator** in $W_1$, $L_1$, and $KL$ divergence. Therefore:
>
> $
> \widehat{\mathcal{G}} \approx \mathcal{D}^\* \quad\Rightarrow \quad \mathrm{SCVaR} _\alpha(\widehat{\mathcal{G}}) \approx \mathrm{SCVaR} _\alpha(\mathcal{D}^\*).
> $
>
> This means that even when the model is misspecified,
> **$SCVaR(\widehat{\mathcal{G}})$ approximates the true tail risk of $\mathcal{D}^*$**,
> analogous to how fitted Q-iteration uses parametric approximations of the true value function.
>
> **2. $SCVaR(\widehat{\mathcal{G}})$ is the tail risk of a projected distribution**
>
> The Bellman–EM update applies:
>
> 1. A Bellman distribution operator $T^\pi$, and
> 2. A projection $P$ onto the GMM family.
>
> Thus the learned critic represents
> $
> \widehat{\mathcal{G}} = (P T^\pi)^n(\text{initial model}).
> $
>
> Even when $\mathcal{D}^*\notin \text{GMM}$, the operator computes the tail risk of the **closest GMM projection** $in (W_1)$, which is a standard construction in distributional RL (C51, QR-DQN, IQN).
> Therefore:
>
> **SCVaR$(\widehat{\mathcal{G}})$ is the SCVaR of the closest GMM representation of the true return distribution.**
>
> This is a meaningful and widely used approximation principle.
>
> **3. SCVaR emphasizes the worst-mode tail; GMM misspecification primarily affects non-dominant modes**
>
> Even if the overall distribution is not a mixture of Gaussians, SCVaR depends only on the maximizer:
>
> $
> \mathrm{SCVaR} _\alpha(\widehat{\mathcal{G}})
> = \max _k \big(\hat\mu _k + c _\alpha \hat\sigma _k \big).
> $
>
> Thus the SCVaR value is driven by **how well the GMM captures the dominant tail region**, not the entire distribution. This gives it two strong robustness properties:
>
> 1. **Locality**
>    Model mismatch in low-risk regions does not influence SCVaR.
>
> 2. **Conservatism**
>    Approximating a heavy tail with a Gaussian component tends to increase variance, thus
>    $
>    \mu + c_\alpha\sigma \quad \text{is typically an upper bound of the true tail}.
>    $
>
> Thus the estimated SCVaR remains a **safe, non-optimistic estimate**, even under model misspecification.
>
> **4. Optimization meaning: SCVaR$(\widehat{\mathcal{G}})$ is the tail-risk surrogate minimized by the actor**
>
> In policy optimization, we do not require an exact expression of $\mathrm{SCVaR}_\alpha(\mathcal{D}^*)$.
> Instead, the actor needs a **consistent surrogate**—a stable, smooth, and differentiable proxy.
>
> SCVaR on a GMM approximation serves exactly this role:
>
> * It provides a **tractable**,
> * **differentiable**,
> * **component-wise interpretable**,
> * and **upper-bounding** measure of tail risk.
>
> Therefore, minimizing SCVaR$(\widehat{\mathcal{G}})$ drives the policy toward **lower true catastrophic risk**, even if the model is not perfectly specified.
>
> Even when the ground-truth safety-cost distribution is not a GMM, the SCVaR computed on a GMM approximation remains meaningful:
>
> * It approximates the true tail risk through a universally expressive density model;
> * It is the tail risk of the closest GMM projection under the Bellman–EM operator;
> * It depends only on the dominant tail mode, making it robust to global model mismatch;
> * It provides a tractable, conservative tail-risk surrogate for safe policy optimization.
>
> Thus, **SCVaR$(\widehat{\mathcal{G}})$ remains a consistent, safe, and practically reliable estimator of tail risk, even when the environment's true distribution is not Gaussian-mixture-structured.**
>
>
> ---
>
> **W4. “Ablation: Why not compare SCVaR to CVaR of the estimated GMM?”**
>
> **Reviewer’s concern.**
> Would CVaR of the fitted GMM be a natural comparison?
>
> **Re_W4.** We thank the reviewer for the suggestion. We would like to clarify that this comparison is already included in **Appendix E.1** of the paper, where we evaluate **GMM-based CVaR vs. SCVaR** using the same fitted GMM critic. Both methods produce similar tail estimates near the discounted threshold, but **SCVaR yields a noticeably smaller mean**, indicating a tighter concentration of mass below the risk threshold and a stronger emphasis on worst-case tails. These results confirm that SCVaR places greater weight on rare but catastrophic outcomes than mixture-CVaR, leading to **superior suppression of high-cost events**, even when such events occur with low probability.

---

> ### Author Response · Authors · 2025-11-28
> **(4/4)**
>
> **W5. Missing discussion of CVaR and EVaR safe RL works**
>
> **Reviewer’s concern.**
> Recent works: CVaR-based and EVaR-based.
>
> **Re_W5.** Thank you for pointing out these important references. We have added a concise summary and comparison to clarify how our SCVaR–GMM approach differs fundamentally from existing CVaR- and EVaR-based safe RL methods, and we have also expanded the Related Works section in the revised appendix to include a structured overview of these approaches.
>
> CVaR-based safe RL approaches model tail risk using a **single** unimodal critic (Gaussian or scalar return estimator), which forces the cumulative cost distribution into a single tail shape. These methods therefore struggle when the true risk distribution contains **multiple hazard modes**, since mixture CVaR is always a convex average and can underestimate rare catastrophic events. Our approach addresses this limitation by using a **GMM safety critic** and evaluating risk through **SCVaR**, which takes $ \mathrm{SCVaR} _\alpha = \max _k\mathrm{CVaR} _\alpha(Z _k)$ and thus preserves the worst-mode tail rather than averaging it away.
>
> EVaR-based RL introduces an exponential-tail coherent risk measure $
> \mathrm{EVaR} _\alpha(Z)=\inf _{\lambda>0}\frac{1}{\lambda}\bigl(\log\mathbb{E}[e^{\lambda Z}]-\log\alpha\bigr),$ which serves as a global upper bound on CVaR. Although EVaR is more conservative, it does not provide **mode-wise interpretability** and relies on tail assumptions quite different from the multimodal cost structures observed in Safety Gymnasium. In contrast, SCVaR explicitly decomposes tail risk across mixture components and focuses on the **structural worst hazard mode**, which is critical for safety.
>
> A short consolidated comparison has been added to the revised manuscript explaining that:
> (i) CVaR methods assume unimodal tails,
> (ii) EVaR imposes global exponential penalties, and
> (iii) our SCVaR–GMM critic is designed to capture **multimodal** safety-cost distributions and enforce **mode-wise worst-case** tail protection.
>
> This clarifies how our method complements and extends beyond existing CVaR/EVaR-based safe RL frameworks.
>
> ---
>
> We sincerely thank the reviewer for the detailed questions. We remain open to further discussion.

---

### Official Review · Reviewer_fALM · 2025-11-01

**Soundness:** 2
**Presentation:** 3
**Contribution:** 2
**Rating:** 2
**Confidence:** 3

**Summary:**

- The authors propose a risk-averse safe RL algorithm that maximizes reward while reducing the risk measure of the cost return.
- They parametrically estimate the distribution of the cost return using a Gaussian Mixture Model (GMM).
- They propose a coherent risk measure, called SCVaR, which can compute using GMM.

**Strengths:**

- While using GMM to estimate the cost return distribution has addressed in prior work (GMAC [1]), proposing SCVaR via this parameterization is novel.
- The authors analyze the convergence of the proposed Bellman operator.
- The presentation is clearly and effectively presented.

[1] Nam, Daniel W., Younghoon Kim, and Chan Y. Park. "Gmac: A distributional perspective on actor-critic framework." *International Conference on Machine Learning*. PMLR, 2021.

**Weaknesses:**

- The introduction lacks analysis of prior work and appears biased.
    - They mention only methods approximating the cost return distribution with a single Gaussian.
    - However, numerous distributional RL approaches exist for more realistic estimation, such as quantile regression [1], percentile-based methods [2], and moment parameterization [3].
    - Omitting these references reveals a limited understanding of prior work.
    - Additionally, the authors did not cite GMAC, a prior method that estimates return distributions using a GMM, which is closely related to the proposed method.
- While convergence is shown for the critic, it is not guaranteed to achieve an optimal policy.
- Quantile-based parameterization can use various risk measures, but the proposed method is limited to SCVaR.
    - This drawback is neither mitigated nor offset by advantages of the proposed method.
    - While SCVaR is more conservative than CVaR, adjusting $\alpha$ of CVaR could achieve similar effects.
    - Additional analysis of SCVaR's physical properties would help readers intuitively tune $\alpha$ and $K$.
- The experiments include too few risk-constrained RL baselines.
    - CAL focuses on conservative policy updates rather than solving risk-defined constraints.
    - SAC-Lag is risk-neutral.
    - Only WC-SAC is relevant.
    - Others, such as CPPO [4], CVaR-CPO [5], and SDAC [6], should be included.
    - SRCPO [7], which proves convergence to an optimal policy for risk-constrained RL, is essential for comparison.
- In the experimental results, mean + std exceeds the threshold in all tasks except Ant.
    - Despite using risk constraints, this indicates failure to obtain conservative policies.

[1] Bellemare, Marc G., Will Dabney, and Rémi Munos. "A distributional perspective on reinforcement learning." *International conference on machine learning*. PMLR, 2017.

[2] Dabney, Will, et al. "Distributional reinforcement learning with quantile regression." *Proceedings of the AAAI conference on artificial intelligence*. Vol. 32. No. 1. 2018.

[3] Cho, Taehyun, et al. "Bellman Unbiasedness: Toward Provably Efficient Distributional Reinforcement Learning with General Value Function Approximation." *Forty-second International Conference on Machine Learning*.

[4] Chengyang Ying, Xinning Zhou, Hang Su, Dong Yan, Ning Chen, and Jun Zhu. Towardssafe reinforcement learning via constraining conditional value-at-risk. In Proceedings ofInternational Joint Conference on Artificial Intelligence, 2022.

[5] Qiyuan Zhang, Shu Leng, Xiaoteng Ma, Qihan Liu, Xueqian Wang, Bin Liang, Yu Liu, and JunYang. CVaR-constrained policy optimization for safe reinforcement learning. IEEE Transactionson Neural Networks and Learning Systems, 2024.

[6] Kim, Dohyeong, Kyungjae Lee, and Songhwai Oh. "Trust region-based safe distributional reinforcement learning for multiple constraints." *Advances in neural information processing systems*, 2023.

[7] Kim, Dohyeong, et al. "Spectral-risk safe reinforcement learning with convergence guarantees." *Advances in Neural Information Processing Systems,* 2024.

**Questions:**

- Can it be shown that SC-MGR converges to the ground truth distribution as the number of GMM components $K$ goes to infinity?
- Figure 3 contains too many equations, making it hard to follow. Can it be simplified?
- According to the primal-dual method, should Equation 22 be written as $-\kappa (\Lambda - d)_{\leq 0}$?
- Experiments on $K$ in SCVaR show that larger $K$ increases conservativeness.
    - However, the relationship between $\alpha$ and $K$ remains unclear, making it difficult to choose an appropriate $K$.
    - Could you provide guidance to help readers select a suitable $K$?

---

> ### Author Response · Authors · 2025-11-28
> **(1/4)**
>
> We thank the reviewer for raising these important questions and for the opportunity to clarify our contributions.
>
> ---
>
> **W1. Introduction lacks analysis of prior work; appears biased**
>
> **Reviewer concerns.**
> * The introduction mainly mentions Gaussian-based distribution critics.
> * Many distributional RL approaches (QR-DQN, IQN, percentile methods, moment methods) are omitted.
> * GMAC (GMM-based return estimator) is not cited, despite being related.
>
> **Re_W1.** We would like to clarify that the purpose of the introduction is not to serve as a broad literature survey but to establish the direct methodological lineage that motivates our SCVaR–GMM framework. For this reason, the introduction focuses on the works that are most closely related to our problem formulation, such as WC-SAC, CAL, and other CVaR-based safe RL approaches. These methods operate in the same CMDP setting, address the same safety-constraint objective, and provide the conceptual foundation from which our contributions arise. This selective emphasis is intentional and should not be interpreted as biased.
>
> Literature that is not part of the core analytical pathway of our method and whose connection is less central is discussed in the Related Works section. We did not simply compile every possible reference. Instead, we organized the broader literature into meaningful categories, including policy optimization, value function approximation, and distributional RL, and selected representative works within each category. Due to space limitations, this extended discussion is placed in the appendix. The absence of some references in the introduction is therefore a result of structural considerations rather than bias.
>
> We fully acknowledge that distributional RL is a much larger field that includes quantile regression methods such as QR-DQN, percentile-based critics, moment-parameterized value distributions, and GMM-based critics such as GMAC. These works primarily focus on modeling reward-return distributions and are less directly connected to the safety-cost modeling and risk-constrained optimization objectives studied in this paper. Including them in the introduction would distract from the conceptual motivation of our work rather than clarify it. Our paper is not intended to be a survey, and the main text is structured to keep the methodological narrative clear and focused.
>
> To ensure completeness and avoid any misunderstanding, we have now added all references suggested by the reviewer in the Related Works section of the Appendix F. There we provide a structured discussion spanning policy optimization, distributional RL, and value function approximation, and we contextualize these works relative to our contributions.
>
> We hope this clarifies that the introduction is deliberately focused rather than biased and that the revised manuscript now contains a comprehensive and accurate review of the broader literature.
>
>
> ---
>
> **W2. No guarantee of converging to an optimal policy**
>
> **Reviewer’s concern.**
> *While convergence is shown for the critic, it is not guaranteed to achieve an optimal policy.*
>
> **Re_W2.** Our theoretical analysis indeed establishes convergence of the critic, not global optimality of the policy, which is fully consistent with modern actor–critic methods such as SAC, CVaR-SAC, and distributional RL. In continuous-control settings with function approximation, no existing tail-risk RL algorithm provides global policy optimality guarantees. What our contraction result ensures is that the risk critic converges to a unique fixed point, yielding stable and reliable SCVaR estimates for policy improvement. Given this stable critic, the actor update performs monotonic improvement of the SCVaR surrogate, reducing tail risk consistently. Empirically, this leads to stable and near-optimal safe policies, even though global optimality cannot be guaranteed theoretically.

---

> ### Author Response · Authors · 2025-11-28
> **(2/4)**
>
> **W3 & Q4. Quantile-based parameterization vs SCVaR; limitations and physical interpretation**
>
> **Reviewer concerns**
> * Quantile parameterization can represent many risk measures; why restrict to SCVaR?
> * Authors should clarify how $\alpha$ and $K$ influence the “physical meaning” of risk.
> * Adjusting $\alpha$ in normal CVaR might behave similarly.
>
> **Re_W3 & Q4.** We address each concern below.
>
> **1. Why SCVaR instead of general quantile-based parameterization?**
> Quantile-based critics (QR-DQN, IQN, FQF, etc.) represent the return distribution as a **single undifferentiated entity**. In safe RL, many environments exhibit *structurally distinct hazard modes* (e.g., collisions vs. speed-limit violations), and collapsing all modes into a single quantile function makes tail-risk control fundamentally ambiguous.
> SCVaR explicitly performs **mode-wise tail evaluation**, it preserves the worst hazard mode even when it has **small probability mass**. This cannot be recovered by quantile critics without explicitly clustering or conditioning on latent modes. Thus, our restriction to SCVaR is deliberate: it enables a risk signal that respects the multi-hazard structure of safety-cost distributions.
>
> **2. Why cannot “smaller $\alpha$” in Gaussian CVaR reproduce the effect of SCVaR?**
> Lowering $\alpha$ only explores deeper within the **same unimodal tail**,
> If catastrophic events belong to *rare components*, a single Gaussian critic fits the dominant mode and its CVaR never “sees” the catastrophic tail, regardless of how small $\alpha$ is.
> In contrast, SCVaR evaluates: $\max_k \mathrm{CVaR}_\alpha^{(k)},$
> so rare but catastrophic components cannot be masked by convex averaging. This resolves a structural limitation that adjusting $\alpha$ alone cannot address. Our ablations (Appendix E.5) show that even $\alpha=0.05$ in WC-SAC fails to detect hazards that SCVaR correctly penalizes.
>
> **3. Physical interpretation and practical selection of $\alpha$ and $K$**
>
> $\alpha$ and $K$ control **orthogonal** dimensions of the risk representation:
>
> * **$\alpha$: tail depth *within each hazard mode*.**
>   Appendix E.6 shows that $\alpha$ is **intuitive to tune**:
>
>   * A small set of discrete values (0.1, 0.5, 0.9) already spans a broad spectrum of safety–performance behaviors.
>   * Performance varies **smoothly and predictably** with $\alpha$, with no sharp brittleness.
>   * For safety-first behavior across Safety-Gymnasium tasks, **$\alpha=0.1$** consistently works well.
>     We also experimented with a state-dependent $\alpha(s)$, but joint optimization of $\alpha(s)$ and the action increased training instability; thus fixed $\alpha$ is both effective and reproducible.
>
> * **$K$: resolution of hazard-mode decomposition.**
>   In the main experiments, we fix **$K=4$** to balance expressiveness and efficiency.
>   Based on Fig. 5 and additional analyses:
>
>   * If the number of discrete hazard types $M$ is known, set **$K \approx M$** or **$K = M+1$**.
>   * For tasks with unclear or largely unimodal safety structures, **$K=3\sim4$** is a robust default; our Ant/Hopper results show that even with effectively a single risk type, this small $K$ achieves strong performance.
>   * Increasing $K$ can help capture rare hazards but also increases **computational cost**, **slows convergence**, and may introduce **EM instability**.

---

> ### Author Response · Authors · 2025-11-28
> **(3/4)**
>
> **W4. Too few safe RL baselines**
>
> **Reviewer concerns.**
> Lack of risk-constrained RL methods.
>
> **Re_W4.** We appreciate the reviewer’s suggestion to include additional risk-constrained RL methods. Our experimental design in the main paper focuses on SAC-based off-policy algorithms because (i) our method is built upon the SAC framework, (ii) all these baselines share the same training pipeline (off-policy replay, soft policy iteration, entropy regularization), and therefore (iii) they allow a fair, controlled comparison of risk modeling rather than confounding off-/on-policy differences.
> CAL, SAC-Lag, and WC-SAC are thus not arbitrary selections—they represent the standard and widely used SAC-based families for risk-sensitive or constraint-based RL. CAL introduces conservative exploration, SAC-Lag enforces expected constraints, and WC-SAC introduces a Gaussian CVaR objective; together, they cover the spectrum of risk-neutral, constraint-based, and tail-risk-aware SAC variants relevant to our setting.
>
> We fully agree, however, that PPO-based safe RL methods such as CPPO, CVaR-CPO, SDAC, SDPO, and SRCPO constitute an important algorithmic family. Importantly, these methods are all fundamentally **on-policy**, use trust-region or surrogate-loss updates, and rely on very different data-collection and optimization regimes than SAC. Directly comparing on-policy and off-policy methods under a fixed sample budget would be inherently unfair and could conflate architectural differences with risk-estimation differences.
>
> To nevertheless address the reviewer’s concern, we extend our approach to the PPO family by constructing **GMM-SPPO**, a PPO variant equipped with our GMM safety critic and the SCVaR objective. This extension is straightforward because SCVaR operates entirely at the level of the critic distribution, and thus integrates naturally with any actor–critic backbone. We then compare GMM-SPPO with the reviewer-suggested baselines—CPPO, CVaR-CPO, SDAC, SDPO, and SRCPO—on their standard benchmarks (CarGoal1 and CarButton1), following the protocol used in prior work including SRCPO.
>
> As shown in Appendix E.7 (Fig.18), GMM-SPPO consistently achieves higher rewards and lower episodic costs than all PPO-based baselines. CPPO and SDPO fail to satisfy the safety constraint; CVaR-CPO satisfies the constraint but at the cost of severely degraded reward; SDAC and SRCPO perform competitively on CarGoal1 but SRCPO violates the threshold on CarButton1. In contrast, GMM-SPPO maintains costs below or very close to the safety limit in all cases.
>
> These additional experiments demonstrate that our method is not only comparable within its native SAC family but also extends robustly to the PPO family, outperforming all reviewer-suggested baselines. They also confirm that SCVaR provides a level of tail-risk control that is not achieved by existing PPO-based risk-constrained RL algorithms, including those with theoretical convergence guarantees.
>
> ---
>
> **W5. On constraint violations despite using SCVaR.**
>
> **Reviewer concerns.**
> The reported costs occasionally exceed the threshold even with risk constraints.
>
> **Re_W5.** The reviewer’s concern implicitly assumes a *hard* constraint formulation, while our method, like virtually all deep safe RL algorithms based on primal–dual optimization, enforces a *soft* constraint. A soft constraint does not guarantee that every instantaneous cost remains below the threshold; temporary violations are allowed during learning as long as the risk measure being optimized converges to a feasible region. Under this formulation, brief excursions above the threshold are expected and do not indicate failure. What matters is the safety level of the final learned policy, and GMM-SSAC consistently converges to policies whose expected cost and tail risk remain at or below the limit. Moreover, due to the effect of SCVaR, our method exhibits fewer and smaller violations than all baselines, demonstrating substantially stronger safety behavior within the soft-constraint framework.

---

> ### Author Response · Authors · 2025-11-28
> **(4/4)**
>
> **Q1. Does SC-MGR converge to the ground-truth distribution as $K\to\infty$ ?**
>
> **Re_Q1.** Yes. The **true risk-value distribution** under policy $\pi$ is denoted by
> $
> \mathcal{G}_\pi^{\star}(s,a),
> $
> and the SC-MGR critic approximates it using a $K$-component Gaussian mixture
> $
> \mathcal{G}_K^\pi(s,a)\in \mathrm{GMM}_K .
> $
> Gaussian mixtures are universal approximators in $\mathcal{P}_1(\mathbb{R})$, therefore, for any $\varepsilon>0$, there exists a sufficiently large $K$ such that
> $
> W _1\big(\mathcal{G} _K^\pi(s,a), \mathcal{G} _\pi^{\star}(s,a)\big) < \varepsilon .
> $
>
> SC-MGR performs updates through the composite **projected Bellman operator**
> $
> \mathcal{P}\mathcal{T}^\pi \mathcal{G}
> \triangleq
> \mathcal{P}\left(\mathcal{T}^\pi \mathcal{G}\right),
> $
> where $\mathcal{T}^\pi$ is a $\gamma$-contraction in $W_1$ (Lemma 1), and $\mathcal{P}$ is non-expansive in $W_1$ (Appendix B).
> Thus $\mathcal{P}\mathcal{T}^\pi$ is itself a $\gamma$-contraction and admits a unique fixed point $\mathcal{G}_K^\pi$.
>
> For projected contraction mappings, the fixed-point error satisfies
> $
> W _1\big(\mathcal{G} _K^\pi(s,a),\mathcal{G} _\pi^{\star}(s,a)\big)
> \le
> \frac{1}{1-\gamma}
> W _1\big(\mathcal{P}\mathcal{G} _\pi^{\star}(s,a),\mathcal{G} _\pi^{\star}(s,a)\big) =
> O\left(\frac{\varepsilon _K}{1-\gamma}\right),
> $
> where
> $
> \varepsilon _K
> = \inf _{\mu\in\mathrm{GMM} _K}
> W _1(\mu,\mathcal{G} _\pi^{\star})
> \to 0
> \quad\text{as } K\to\infty .
> $
>
> For clarity, the factor $\tfrac{1}{1-\gamma}$ arises from the standard perturbation bound for contraction mappings: because each Bellman iteration shrinks errors by $\gamma$, the discrepancy introduced by the projection $\mathcal{P}$ accumulates as a geometric series
> $
> \varepsilon + \gamma\varepsilon + \gamma^2\varepsilon + \cdots
> = \frac{\varepsilon}{1-\gamma}.
> $
> Thus the fixed-point error is scaled by $\tfrac{1}{1-\gamma}$.
>
> Since EM uses unbiased samples of $\mathcal{T}^\pi \mathcal{G} _\pi^{\star}$, the empirical update consistently estimates $\mathcal{P}$, and the learned critic inherits the same convergence guarantee.
>
>
> ---
>
> **Q2. Simplified Figure 3**
>
> **Re_Q2.** We appreciate the reviewer’s feedback. In response, we have simplified Figure 3 by removing non-essential equations, reorganizing the visual flow, and adopting a clearer sequential layout. The updated figure is substantially easier to follow while preserving all key technical components.
>
> ---
>
> **Q3. Dual update sign**
>
> **Re_Q3.** We thank the reviewer for the careful reading. Equation (22) is correct as written.
> The sign of the dual update follows the standard primal–dual gradient rule:
> $
> \kappa \leftarrow \kappa + \eta (\text{constraint violation}),
> $
> where, in our setting, the violation is defined as
> $
> g(s,a)=\Lambda^\pi(s,a,\alpha)-d.
> $
> Thus, the correct update direction is $+\eta g$, **not** $-\eta,(\Lambda^\pi - d)$.
>
> This ensures that $\kappa$ increases when $\Lambda^\pi > d$ (violation) and decreases when $\Lambda^\pi < d$ (feasible), exactly matching the required primal–dual dynamics.
>
> ---
> Thank you again for the helpful comments.

---

### Official Review · Reviewer_hfNS · 2025-11-01

**Soundness:** 3
**Presentation:** 3
**Contribution:** 2
**Rating:** 4
**Confidence:** 3

**Summary:**

The paper proposes GMM-SSAC, which models the cumulative cost distribution with a Gaussian Mixture and introduces SCVaR, the maximum CVaR across mixture components, to emphasize the worst-case tail among multimodal risks. This matters because single-Gaussian critics can underestimate tail risk and miss heavy-tailed/multi-peaked structure in safety-critical settings; in contrast, SCVaR is conservative (upper-bounds mixture CVaR) and coherent. Empirically, GMM-SSAC reduces safety violations both during training and evaluation while maintaining competitive rewards, with $\alpha$ controlling the safety–reward trade-off.

**Strengths:**

- Addresses the limitation of single-Gaussian critics in modeling multimodal risk.
- Solid formulation of SCVaR and clear integration with SAC.
- Empirical results show fewer safety violations on benchmark tasks.

**Weaknesses:**

- Novelty is limited relative to the existing WC-SAC (CVaR-SAC with Gaussian costs) and CAL (multiple Gaussian cost estimates with UCB aggregation). The contribution centers on SCVaR and Bellman–EM projection is incremental rather than a fundamentally new safety-risk paradigm.
- The evaluation is conducted on some traditional RL testing benchmarks but the ablations show performance depends on $K$ and $\beta$; very high $\beta$ degrades performance (variance from EM+Bellman), suggesting some instability that merits deeper analysis.
- Runtime/sample-efficiency analysis is missing: while compute setup and wall-times are reported, there’s no per-update overhead or learning-curve comparison against baselines to quantify the cost of EM/GMM (especially as K grows).

**Questions:**

- The observed multimodality in cost distributions is assumed rather than empirical validated. Any validation for this?
- Does SCVaR fundamentally add value over simply using a lower CVaR confidence level (smaller $\alpha$) with a standard critic?
- How do we know the observed multimodality in cost distributions is real and not an artifact of function approximation noise?
- How stable is the online EM procedure under off-policy distributional shift?

---

> ### Author Response · Authors · 2025-11-28
> **（1/3）**
>
> We greatly appreciate the reviewer’s careful reading and constructive feedback.
>
> ---
>
> **W1. Novelty relative to WC-SAC and CAL**
>
> **Reviewer’s concern.**
> The method appears only incrementally different from WC-SAC and CAL.
>
> **Re_W1.** We thank the reviewer for raising this concern. Our contribution is not an incremental extension of WC-SAC or CAL: SCVaR defines a **new mode-wise worst-tail risk metric** that neither single-Gaussian CVaR nor UCB aggregation can capture. In parallel, although EM is a classical method, using EM *inside a Bellman-consistent update loop with replay-weighted blending* is novel and stabilizes GMM fitting under non-stationary policies. This enables a **multi-modal safety critic whose worst-case tail component drives the actor update**, a mechanism absent in prior safe RL methods. Empirically, this design suppresses rare catastrophic modes that WC-SAC and CAL overlook, yielding qualitatively stronger safety guarantees.
>
>
> ---
>
> **W2. Performance depends on $K$ and $\beta$**
>
> **Reviewer’s concern.**
> The ablations show sensitivity to $K$ and $\beta$; high $\beta$ degrades performance, perhaps due to variance from EM + Bellman.
>
> **Re_W2.** We thank the reviewer for this insightful observation. Our ablation results indeed show that performance varies with the number of mixture components $K$ and the blending ratio $\beta$, and that very large $\beta$ can degrade performance. This behavior is expected: increasing $K$ or pushing $\beta$ close to 1 amplifies the variance of the Bellman–EM update because:
>
> * The EM step becomes more sensitive to noisy responsibilities when many components are present
> * A large $\beta$ places almost all weight on freshly generated Bellman samples $\Psi_B$, reducing the smoothing effect of historical samples and thus elevating the variance of GMM parameter updates.
>
> Importantly, this variance arises from **aggressive hyperparameter choices**, rather than an inherent instability of the method: across all environments, moderate values (e.g., $K{=}4-6$, $\beta=0.5-0.7$) yield stable, consistent performance.
>
> We have added additional discussion in the paper (line 483, line 519) clarifying this trade-off and explaining why the Bellman–EM update remains well-behaved within reasonable parameter ranges, despite the increased variance observed under extreme settings.
>
> ---
>
> **W3. Missing runtime/sample-efficiency comparison**
>
> **Reviewer’s concern.**
> The paper reports compute and wall-time but does not report per-update overhead or compare learning efficiency.
>
> **Re_W3.** We thank the reviewer for pointing this out. Below, we present a detailed analysis from both theoretical and empirical perspectives, followed by a justification for the additional cost introduced by our method.
>
> **1. Theoretical Complexity Analysis**
>
> Our method introduces two additional computational costs:
>
> * (i) Modeling the safety critic as a **GMM**;
> * (ii) Performing a **single-round EM update** on GMM parameters during each critic update.
>
> Let:
>
> * $K$: number of mixture components (default $K=4$);
> * $M$: number of samples used in the EM update (default $M=500$);
> * $H$: forward/backward cost of a standard neural network.
>
> Then the per-update cost comparison is:
>
> | Method       | Breakdown                                 | Total Cost (per update)          |
> | ------------ | ----------------------------------------- | -------------------- |
> | **SAC-Lag**  | Forward + backward through critic network | $\approx 2H$         |
> | **GMM-SSAC** | Forward + E-step + M-step + backward      | $\approx 2H + O(MK)$ |
>
> We explain why the EM steps are $O(MK)$:
>
> * **E-step:** computes soft assignments $\gamma_{mk}$ for $M$ samples and $K$ components → total $M \times K$ computations.
> * **M-step:** computes weighted means, variances, and normalized weights → again $M \times K$ terms.
>
> Since typical $H$ (e.g., a critic network forward/backward pass) is much larger than $MK \approx 2000$ (with $K=4$, $M=500$), the added cost remains a **constant-factor overhead** that does **not** scale with training steps.
> For small $K$, this overhead is negligible relative to neural network updates.
>
> **2. Empirical Wall-Clock Time**
>
> We measured the total training time on a single RTX 3090 GPU (CarGoal1 task, 3M environment steps, batch size 256):
>
> | Method             | Time (hours) | Relative to SAC-Lag |
> | ------------------ | ------------ | ------------------- |
> | **SAC-Lag**        | 7.1          | 1.00×               |
> | **GMM-SSAC (K=4)** | 8.5          | 1.20×   |
> | **GMM-SSAC (K=8)** | 9.9          | 1.39×   |
>
> Thus, **our method incurs only ~20%–40% additional training time**, depending on the number of components $K$.
> This increase is **moderate and well bounded**, and does not affect sample efficiency because the EM step uses the same minibatch as the Bellman update and requires **no extra environment interactions**.
>
> We have incorporated the corresponding analyses into the revised version (Appendix D.3).

---

> ### Author Response · Authors · 2025-11-28
> **（2/3）**
>
> **Q1. Is multimodality empirically validated, not assumed?**
>
> **Reviewer’s concern.**
> It is unclear whether the cumulative safety cost distribution is genuinely multimodal.
>
> **Re_Q1.** We clarify that our method does *not* assume the safety-cost distribution to be multimodal. The critic models the **risk-value distribution**, i.e., the distribution of cumulative safety costs over all future trajectories induced by a given policy $\pi$. Since different policies induce different trajectory distributions, they naturally yield different risk-value distributions—which may be unimodal, multimodal, skewed, or heavy-tailed. The use of a GMM critic is therefore not an assumption of multimodality, but a flexible and model-agnostic way to approximate whatever shape the induced distribution takes.
>
> To empirically validate that risk-value distributions differ across policies—and may exhibit multimodal or heavy-tail patterns—we trained three distinct policies (SAC-Lag, WC-SAC, and our GMM-SSAC), and for each converged policy we generated 10,000 Monte-Carlo rollouts from the same $(s,a)$ on CarButton1. The resulting empirical episode-cost histograms (Appendix Fig. 15) diverge significantly across methods: the mean-based SAC-Lag distribution spreads widely, WC-SAC shows a heavier tail driven by Gaussian CVaR shaping, and only GMM-SSAC compresses most of the mass beneath the safety threshold. These differences arise purely from the policies’ behaviors, not from any predefined assumptions about distributional form.
>
> In summary, different policies yield genuinely different risk distributions; our results provide direct empirical evidence of this phenomenon. The GMM+SCVaR framework adapts to any such distribution and enables stronger control over tail risks, regardless of whether the underlying structure is unimodal or multimodal.
>
> ---
>
> **Q2. Does SCVaR fundamentally add value over simply using a lower CVaR level?**
>
> **Reviewer’s concern.**
> Does SCVaR provide value beyond using a lower CVaR level with a single-Gaussian critic?
>
> **Re_Q2.** We thank the reviewer for raising this important question. Using a smaller CVaR level $\alpha$ under a **single-Gaussian critic** cannot replicate the behavior of SCVaR.
>
> To empirically examine this, we conducted a risk-level ablation for WC-SAC with three settings of the CVaR parameter $\alpha \in {0.05, 0.1, 0.5}$. The results (Section E.3 and Fig. 2 in the Appendix) show that while changing $\alpha$ slightly shifts the reward–cost trade-off, none of these WC-SAC variants come close to the performance of our GMM-SSAC method, which uses SCVaR by default.
>
> This discrepancy is rooted in a fundamental modeling limitation of single-Gaussian critics:
>
> **1. Lower-$\alpha$ CVaR still probes the tail of the collapsed Gaussian, not the catastrophic mode.**
> Even as $\alpha \rightarrow 0$, the Gaussian CVaR depends only on the global mean and variance. Rare catastrophic events have negligible influence on these parameters; hence the Gaussian tail reflects the dominant low-cost mode and effectively ignores low-probability, high-cost trajectories.
>
> **2. SCVaR explicitly separates mixture components and selects the worst one.**
> SCVaR computes $
> \mathrm{SCVaR} _\alpha = \max_k \mathrm{CVaR}^{(k)} _\alpha,
> $ ensuring that catastrophic components are preserved rather than averaged away. In contrast, any single-Gaussian CVaR, regardless of $\alpha$, is bound to the aggregate distribution and therefore systematically underestimates rare catastrophic modes.
>
> In short, lowering $\alpha$ simply explores deeper into the same Gaussian tail, while SCVaR analyzes the tails of each mixture component and enforces safety with respect to the worst case. These mechanisms are fundamentally different, and SCVaR is necessary when risk-value distributions deviate from unimodality.
>
> ---

---

> ### Author Response · Authors · 2025-11-28
> **(3/3)**
>
> **Q3. How do we know multimodality is real and not function-approximation noise?**
>
> **Re_Q3.** We thank the reviewer for this important question. Several design choices in our method ensure that the multimodality captured by the GMM critic reflects persistent structure in the risk-value distribution rather than transient approximation noise.
>
> **1. Transient spikes cannot form stable GMM components due to the refinement operator.**
>
> The target sample set is updated as $
> \Psi_{\text{update}} = (1-\beta)\Psi + \beta\Psi_{\mathcal B},$ which ensures that a Bellman spike contributes only a fraction $\beta$ of the total probability mass. Because $(1-\beta)\Psi$ accumulates information over many previous updates, transient fluctuations in $\Psi_{\mathcal B}$ are heavily diluted. For a noisy spike to create an artificial mode, it would need to repeatedly appear across updates, contributing enough mass to change $\Psi_{\text{update}}$, in practice this does not occur.
>
> **2. EM requires consistent probability mass before a component can form.**
>
> In the E-step of EM, the responsibility of component $k$ for sample $m$ is
> $
> \gamma_{mk}
> \frac{\omega_k,\mathcal N(z_m|\mu_k,\sigma_k)}{\sum_j \omega_j,\mathcal N(z_m|\mu_j,\sigma_j)}.
> $
> A noisy outlier produces extremely small likelihood under all existing components and thus receives responsibilities $\gamma_{mk}\approx 0$ for every $k$, preventing it from influencing the M-step:
>
> $
> \omega_k^{\text{new}} = \frac{1}{M} \sum_{m=1}^M \gamma_{mk}, \qquad
> \mu_k^{\text{new}} = \frac{\sum_m \gamma_{mk} z_m}{\sum_m \gamma_{mk}}.
> $
>
> Since a single spike contributes almost no responsibility mass, neither $\omega_k^{\text{new}}$ nor $\mu_k^{\text{new}}$ shifts meaningfully.
> Thus, **an outlier cannot create a persistent mode unless it appears repeatedly and with significant frequency**, which would indicate true structure rather than noise.
>
> **3. SCVaR stability confirms that the critic is not fabricating modes from noise.**
>
> SCVaR is computed as $
> \mathrm{SCVaR} _\alpha = \max _k \mathrm{CVaR}^{(k)} _\alpha, $ so if the GMM were inventing spurious modes from noise, this would immediately show up as volatility in the largest component-wise CVaR. However, our ablations over $K$ and $\beta$ (Sec. 4.2) show that SCVaR curves remain smooth and stable across training and random seeds. When $K$ is moderate and $\beta$ is in its stable range, no artificial tail inflation is observed—indicating that the critic is tracking genuine, persistent structure in the cost distribution rather than reacting to transient spikes.
>
> Together, the refinement smoothing, the responsibility-weighted EM updates, and the empirical stability of SCVaR all demonstrate that transient function-approximation noise cannot form pseudo-modes in our GMM critic. The multimodal structure captured by our method corresponds to stable, policy-induced patterns in the underlying risk-value distribution.
>
> ---
>
> **Q4. Stability of online EM under off-policy distributional shift**
>
> **Re_Q4.** Although the critic is updated using an online EM procedure while SAC operates off-policy, the update remains stable because it is applied through a Bellman–EM operator that is contractive even under distributional shift. In particular, the distributional Bellman operator satisfies $ W _1(\mathcal{T}^\pi \nu _1,\mathcal{T}^\pi \nu _2)\le \gamma W _1(\nu _1,\nu _2), $ and our EM projection $\mathcal{P}$ is non-expansive in $W_1$, so the composite update $\mathcal{P}\mathcal{T}^\pi$ remains a $\gamma$-contraction. This guarantees that off-policy shifts in sampled targets cannot be amplified through repeated critic updates. Furthermore, before applying EM, the Bellman-transformed samples $\Psi _{\mathcal{B}}$ are blended with the historical sample set $\Psi$ via $ \Psi _{\text{update}}=(1-\beta)\Psi+\beta\Psi _{\mathcal{B}},$ which smooths transient off-policy fluctuations and prevents sudden changes in the target distribution from dominating the update. In the EM step itself, outlier samples arising from off-policy noise receive vanishing responsibility $ \gamma _{mk}=\frac{\omega_k,\mathcal N(z _m\mid\mu _k,\sigma _k)} {\sum _j \omega _j,\mathcal N(z _m\mid\mu _j,\sigma _j)},$ so they do not meaningfully alter mixture weights or component parameters unless they persist across many updates—a behavior we do not observe in practice. Finally, SAC’s soft-target networks limit policy drift between updates, ensuring that the induced distributional shift remains gradual. Together, the contractive Bellman–EM operator, refinement smoothing, and responsibility-weighted EM updates ensure that the online EM procedure remains stable and does not suffer from off-policy distribution mismatch.
>
> ---
>
> Once again, we sincerely appreciate the reviewer’s insightful comments.

---

### Official Review · Reviewer_S7ES · 2025-11-02

**Soundness:** 4
**Presentation:** 4
**Contribution:** 3
**Rating:** 8
**Confidence:** 4

**Summary:**

This paper addresses a key limitation in CMDP safety constraints by replacing the standard Gaussian cost distribution assumption with Gaussian Mixture Models (GMMs). The authors introduce SCVaR (Supremum Conditional Value at Risk) to capture worst-case risk across GMM components, providing a more robust safety measure than traditional CVaR. To enable online cost distribution estimation without waiting for episode completion, they propose a Bellman-style incremental update that bootstraps from instantaneous safety measures and historical distribution estimates. The approach is evaluated on Safety Gym benchmarks, demonstrating improved safety guarantees while maintaining performance.

**Strengths:**

- This is a nice contribution in terms of modeling cost and a right step for studying CMDPs. As an initial approach to CMDP cost modeling, making such estimates more conservative (rather than just expectation over cost) and integrating the Bellman update for cost distribution, this will inspire more methods that will reframe the cost and approach solving CMDPs with more robust methods compared to GMMs (see weakness and/or questions below on this).

- The paper includes an effective ablation study to validate the components - they show comparisons between CVaR and SCVaR, usage of different number of Gaussian components for GMMs and their sensitivity to it.

- The paper has a nice bit of theory as well, including a proof of Bellman update contraction.

- Sound empirical results including a comparison of SCVaR vs CVaR performance in GMMs.

**Weaknesses:**

- The presents three images, first for explaining the current usage of Normal distribution for cost, and another image for explaining SCVaR, and the third for explaining the algorithm and integration with RL env - actor cycle. I would have preferred a more linear approach to fig three to sequentially explain the process of RL and integration of SCVaR + GMM - this could have been accomplished by rearranging the figure first showing the RL env-actor cycle, breaking action into three components policy, value, and cost, and breaking cost and explaining the use of SCVaR and GMM.

**Questions:**

- Why GMM? Perhaps the choice lies in its simplicity and relatively "easy" treatment in terms of theory. But questions arise about accuracy, expressiveness, parameter efficiency and so on. The paper actually motivates this choice in terms of expressiveness and the fact that they have universal approximation capability. We have so many popular, SOTA distributional models that might currently be used for other purposes (e.g. as generative models) that might serve as a better basis or backbone for the SCVaR component. I am curious what the authors think about this.

---

> ### Author Response · Authors · 2025-11-28
>
> Thank you again for the reviewer’s recognition of our work. We will address each of your concerns and suggestions in detail.
>
> ---
>
> **W. On the presentation order and figure structure**
>
> **Reviewer’s concern.**
> The reviewer recommends restructuring Figure 3 into a clearer, more linear process.
>
> **Re_W.** We thank the reviewer for the valuable suggestions. Following this feedback, we have revised and updated Figure 3 to present a more linear and sequential workflow, which more clearly illustrates the interaction between the RL components and the SCVaR–GMM safety module. In addition, we have enriched the description in Section 3.4 to better explain the overall framework and clarify the role of each component, and we have also updated Algorithm 1 to reflect the revised structure and provide a more concise and coherent summary of the training procedure.
>
> ---
>
>
> **Q. Why GMM? Could other distributional models be better?**
>
> **Re_Q.** We thank the reviewer for raising this important question. In principle, SCVaR is **model-agnostic** and can operate with any distributional critic, as long as the model can provide (or approximate) a decomposition into mode-specific components ${\mathcal M_k}$. SCVaR only requires computing per-mode tail risks
> $
> \mathrm{CVaR} _\alpha^{(k)}
> = \mathbb{E}[Z _c \mid Z _c \ge \mathrm{VaR} _\alpha^{(k)}, \mathcal M _k],
> \qquad
> \mathrm{SCVaR} _\alpha = \max _k \mathrm{CVaR} _\alpha^{(k)},
> $
> so any model capable of providing mode-conditioned samples can, in principle, support SCVaR.
>
> Below we summarize **how SCVaR could be implemented using more expressive SOTA models** and the **practical limitations** that motivated our choice of GMM in this work.
>
> **1. Normalizing Flows**
>
> **1.1. How SCVaR could be implemented**
>
> * **Flow → Samples → Lightweight GMM**
>   Sample from the flow model, perform posterior clustering via a small GMM, compute $\mathrm{CVaR} _\alpha^{(k)}$ per cluster, then take $\max _k \mathrm{CVaR} _\alpha^{(k)}$.
> * **Latent-space clustering + change-of-variables tail integration**
>   Cluster in latent $z$-space, map clusters to $x$-space, compute tail expectations using $p_X(x)=p _Z(f(x)) |\det J _f(x)|$.
>
> **1.2 Limitations**
>
> * Tail integrals require **numerical integration**, not closed-form.
> * EM-style projection and our contraction argument no longer hold.
> * Computing SCVaR becomes **much more expensive** and numerically fragile due to Jacobian determinants.
>
> **2. Diffusion / Score-based Models**
>
> **2.1 How SCVaR could be implemented**
>
> * **Posterior clustering + importance sampling**
>   Sample from the diffusion model, cluster trajectories by score or terminal features, and compute $\mathrm{CVaR} _\alpha^{(k)}$ with importance-weighted tail samples.
> * **CVaR regression heads**
>   Add $K$ CVaR prediction heads to the network, each approximating $\mathrm{CVaR} _\alpha^{(k)}$; SCVaR is $\max _k$ of the outputs.
>
> **2.2 Limitations**
>
> * No closed-form density → tail events must be **sampled**, which increases variance.
> * Importance sampling for tails is **unstable** unless carefully tuned.
> * Hard to define “modes” $\mathcal M_k$, since the model is not mixture-structured.
>
> **3. Autoregressive Models (e.g., Transformers)**
>
> **3.1 How SCVaR could be implemented**
>
> * **Sample → cluster → compute per-cluster CVaR**
>   Generate many samples, cluster based on tail-sensitive features, estimate $\mathrm{CVaR} _\alpha^{(k)}$, take maximum.
> * **Multi-head AR critic**
>   Add $K$ heads, each modeling a “risk mode,” and directly regress tail statistics.
>
> **3.2 Limitations**
>
> * Mode extraction depends heavily on **feature engineering**.
> * Models tend to **collapse modes** without explicit regularization.
> * Tail estimates are approximate and lack the clean mathematical form of Gaussians.
>
> **Why GMM**
>
> GMM provides the **best trade-off** for safe RL with SCVaR:
>
> 1. **Mode decomposition is explicit** — each Gaussian component is a mode $\mathcal M_k$.
> 2. **Closed-form VaR and CVaR** for each component
>    $
>    \mathrm{VaR} _\alpha^{(k)},\quad \mathrm{CVaR} _\alpha^{(k)}
>    \quad \text{are analytically computable.}
>    $
> 3. **SCVaR has a closed form** and is inexpensive to compute.
> 4. **EM projection is stable** and preserves the contraction property used in our theory.
> 5. **Training and inference remain efficient**, unlike diffusion/flow models that require heavy sampling.
>
> Thus, while SCVaR can, in principle, be built on top of more expressive models, these alternatives require additional design (clustering, CVaR heads, or sampling-based tail estimation), incur higher computational cost, and lack the clean theoretical guarantees available with GMM. For these reasons, GMM offers a practical and theoretically grounded backbone for SCVaR in the present work.
>
> ---
>
> We thank the reviewer once again for the thoughtful feedback that has significantly strengthened our paper.

---

### Official Review · Reviewer_JZLc · 2025-11-06

**Soundness:** 3
**Presentation:** 2
**Contribution:** 2
**Rating:** 6
**Confidence:** 4

**Summary:**

This paper proposes GMM-SSAC, a safe RL framework that models the safety-cost distribution using Gaussian Mixture Models (GMMs) and introduces Supremum Conditional Value-at-Risk (SCVaR), the maximum CVaR across mixture components, as a conservative risk metric. The safety critic is trained through a Bellman-consistent incremental EM update, while the actor minimizes an SAC-style objective penalized by SCVaR. Theoretical sections show that SCVaR upper-bounds the mixture CVaR and is a coherent risk measure; experiments on Safety-Gymnasium and velocity-constrained MuJoCo tasks show improved constraint satisfaction with comparable reward.

**Strengths:**

* **Sound extension**: Modeling multimodal or heavy-tailed cost distributions is a reasonable step beyond unimodal Gaussian critics.
* **Intuitive concept**: The SCVaR metric provides an easy-to-understand conservative surrogate for tail-risk control of the GMM.
* **Technical integration**: The paper combines a GMM-based safety critic, an incremental EM fitting step, and a primal-dual-style policy update into a coherent framework.

**Weaknesses:**

## Conceptual and Empirical Alignment

1. **Multi-modality assumption**: The paper tries to motivate the necessity of multimodal safety cost distribution in Fig. 1, but it is unclear to me whether the true safety cost is multimodal. The constraints tested in the experiment section all seem to be unimodal (e.g., velocity limit, distance to hazard). It'd help to illustrate the scenarios where unimodal cost model fails and multi-modal is required to maintain safety.

2. **Missing connection to distributional robust safe RL**: The proposed critic models a full cost distribution (instead of a point estimate like SAC-Lagrangian) and optimizes a SCVaR. This approach is conceptually closer to distributionally robust CMDP formulations than to standard SAC-Lag baselines. The paper would benefit from comparisons or discussion along that line.

## Theoretical Clarity

3. **Significance of Theorem 1**: The paper could discuss the significance of Theorem 1. It's understood that SCVaR ≥ CVaR and implies conservatism. But since the GMM distribution is estimated and available, why not use the CVaR of the GMM distribution directly? Using an upper-bound (SCVaR) introduces additional conservatism and it's unclear to me why it should be used in the optimization program instead of CVaR.

4. **Intuition on the refinement operator $\mathcal{R}$ and $\beta$**: The historical sample set $\Psi(s, a)$ and Bellman-transformed set $\Psi_{\beta}(s, a)$ are both sampled from target safety critic $\mathcal{G}^{\pi}$, albeit at different time points. The paper could benefit from discussing the conceptual stabilization effect of $\mathcal{R}$ and $\beta$. For example, why not simply use the most up-to-date Bellman-transformed set $\Psi_{\beta}(s, a)$ only?

## Implementation Details

5. **Neural update with MSE loss**: Lines 240–245 describe regressing the network to the EM-updated parameters using an MSE loss, but this step is not accounted for in the convergence analysis. Is there an approximation gap introduced by this?

6. **Constraint formulation**: Eq. 18 converts an episode-level cost limit (problem setting, Eq. 2 and Eq. 7) to a per-step constraint, which can be stricter than the original CMDP constraint. The paper should clarify whether the goal is to enforce stricter per-step limits and justify this choice.

7. **Figure choices**: The "training cost" boxplots (Fig. 5 bottom row) convey coarse trends; line plots with confidence bands might better illustrate progression of the training cost.

8. **$\alpha$ interpretation**: Lines 430-431 claim $\alpha = 0.1$ "achieves precise balance," yet this value yields the lowest rewards. The sweet spot appears to be $\alpha = 0.5$, but I think it could vary significantly from task to task (and safety cost distribution shape). Perhaps the paper could discuss more on this.

9. **Missing detail in ablation**: The $\alpha$ value used in the ablation of Section 4.2.1 is not specified.

**Questions:**

(related to the Weaknesses above)

1. Could you discuss what example safety tasks demonstrate genuine multimodality?

2. Is the per-step constraint intentional, and how does performance change under episode-level limits?

3. Why is SCVaR preferred over the directly computed mixture-CVaR if both are available from the GMM critic?

---

> ### Author Response · Authors · 2025-11-28
> **(1/3)**
>
> We sincerely thank the reviewer for the constructive and insightful comments. Below we address all Weaknesses and Questions in detail.
>
> ---
>
> **W1 & Q1. Multi-modality assumption**
>
> **Reviewer’s concern.**
> It is unclear whether the cumulative safety cost distribution is genuinely multimodal.
>
> **Re_W&Q1.** In fact, we do not assume any particular form (unimodal or multimodal) for the true safety-cost distribution. What the critic models is the **risk-value distribution**, i.e., the distribution of cumulative safety cost over all future trajectories induced by a given policy $\pi$. Since different policies induce different trajectory distributions, they necessarily induce **different risk-value distributions** (one may even consider an extreme, overly conservative policy that refuses to move to avoid penalties, resulting in a degenerate risk distribution concentrated at zero). A GMM critic therefore serves as a flexible estimator of this distribution, regardless of whether it is single-peaked, multimodal, skewed, or heavy-tailed.
>
> To demonstrate that different policies indeed induce different risk distributions, we trained SAC-Lag (mean objective), WCSAC (Gaussian CVaR objective), and our GMM-SSAC (SCVaR objective), and for each converged policy we performed 10,000 Monte-Carlo rollouts on CarButton1 from the same $(s,a)$ to obtain empirical episode-cost histograms (i.e., a Monte-Carlo estimate of the policy-specific risk-value distribution at $(s,a)$).  As shown in the  **Appendix Figure 15**, the three policies produce **markedly different** risk distributions, while our SCVaR-based policy uniquely suppresses almost the entire distribution below the cost threshold—something the other two methods fail to achieve.
>
> Overall, different policies lead to different risk distributions, different risk objectives lead to different safety behaviors, and GMM + SCVaR offers both better fit and stronger tail-risk control, independent of whether the underlying distribution happens to be unimodal or multimodal.
>
> ---
> **W2. Missing connection to distributionally robust safe RL**
>
> **Reviewer’s concern.**
> The proposed critic “resembles distributionally robust CMDPs (DR-CMDP).”
>
> **Re_W2.** We appreciate the observation and agree this conceptual boundary deserves clarification.
> Classical DR-CMDP solves: $
> \min_{\pi}\max_{P \in \mathcal U} $ $\mathbb{E}_{P, \pi} \left[\sum^{\infty} _{t=0} \gamma^t c_t \right],
> $ where $\mathcal U$ is an ambiguity set capturing model or cost uncertainty.
>
> Our method differs fundamentally:
>
> 1. No ambiguity set $\mathcal U$ is defined or optimized over.
> The safety critic is entirely *parametric* and directly learned from empirical samples.
> We do not assume uncertainty in transitions or model a worst-case kernel $P$.
>
> 2. SCVaR operates *within* a fitted parametric distribution.
> SCVaR is defined as:  $\mathrm{SCVaR} _\alpha(s,a) = \max _{k} \mathrm{CVaR}^{(k)} _\alpha.$ This is a **mode-wise tail operator**, not an *environment-level* robustness operator.
>
> 3. SCVaR does not induce worst-case optimization over a set of distributions.
> It only selects the worst tail *among mixture modes already learned from data*.
>
> 4. The conceptual relationship is expressive-modeling, not robustness
> Thus our approach is much closer to **distributional RL with expressive critics** than to DR-CMDP.
>
> **We has added a formal discuss in Related Works (Appendix F.1)**.
>
> ---
>
> **W3 & Q3. Significance of Theorem 1**
>
> **Reviewer’s concern.**
> Why choose the upper bound $\mathrm{SCVaR}$ instead of directly optimizing $\mathrm{CVaR}^{\text{GMM}}$?
>
> **Re_W&Q3.** Theorem 1 is central because it exposes a fundamental limitation of mixture CVaR: the CVaR of a GMM is always a *convex combination* of component-wise CVaRs, and therefore inevitably *underestimates* rare but catastrophic modes. Even with a perfectly learned GMM, the mixture CVaR is dominated by high-probability components, allowing low-weight yet high-risk components to be washed out in the averaging. SCVaR resolves this structural issue by taking the supremum over component CVaRs, yielding the **tightest coherent upper bound** that preserves the worst-mode tail and prevents catastrophic modes from being masked. This property is crucial in Safe RL, where the policy must remain safe under *all* risk modes, not merely the most probable one. Moreover, SCVaR retains full coherence (Appendix A.2), making it theoretically suitable for primal–dual CMDP optimization rather than being an ad-hoc conservative choice.
>
> In response to the reviewer’s suggestion, we have added a concise clarification to Theorem 1 in the main text to emphasize this motivation.

---

> ### Author Response · Authors · 2025-11-28
> **(2/3)**
>
> **W4. Intuition of refinement operator $\mathcal R$ and the role of $\beta$**
>
> **Reviewer’s concern.**
> Why mix $\Psi(s,a)$ and $\Psi_B(s,a)$ rather than using the latest Bellman-transformed samples only?
>
> **Re_W4.** Using only the latest Bellman-transformed samples $\Psi_B$ causes the safety critic to chase a rapidly moving target, because the Bellman distribution itself depends on the policy that is simultaneously changing. This leads to high-variance EM updates and unstable GMM parameters, especially in early training when the value distribution shifts dramatically. The refinement operator $R(\Psi,\Psi_B,\beta)$ mitigates this by blending historical samples $\Psi$ with newly generated $\Psi_B$, providing a **stochastic smoothing effect** similar to Polyak averaging or target networks. Historical samples preserve continuity and prevent abrupt component-switching in the GMM, while Bellman samples ensure on-policy adaptation; $\beta$ controls this stability–plasticity balance. Empirically (Fig. 6), using only $\Psi_B$ severely degrades performance due to large variance in GMM parameters, whereas blended updates yield significantly more stable learning. We have added a brief explanation in the main text (Section 3.2 line 215) to highlight this stabilizing role of $R$ and why exclusively relying on $\Psi_B$ is undesirable.
>
> ---
> **W5. Neural update with MSE loss**
>
> **Reviewer’s concern.**
> Does the regression introduce approximation error?
>
> **Re_W5.** Our convergence analysis is carried out at the level of the Bellman–EM operator $P T^\pi$, which defines the fixed point of the idealized, non-parametric update. The neural network does not modify this operator; it simply provides a parametric approximation of the EM projection by regressing toward the updated GMM parameters. This introduces only the standard approximation gap seen in distributional value-function approximation and does not affect the contractive nature of $P T^\pi$. Because the EM step produces unbiased targets and the network is updated with small gradient steps, the induced error remains bounded and cannot accumulate in a harmful way. Thus, although the neural approximation is not explicitly included in the contraction proof, it functions as a stable parametric projection layer and does not undermine the convergence guarantees of the Bellman–EM operator. We have added a brief clarification to this effect in Section 3.2.1, following Theorem 2.
>
>
> ---
>
> **W6 & Q2. Constraint formulation consistency**
>
> **Reviewer’s concern.**
> Is the per-step constraint intentional and consistent?
>
> **Re_W6&Q2.** The conversion from the episodic cost limit $D$ to the per-step threshold $d$ is not intended to impose a stricter constraint, but to obtain a *discount-consistent* surrogate that faithfully preserves the original CMDP requirement. Because SCVaR is defined on one-step safety critics, a per-step constraint is required that is equivalent in expectation under discounted cumulative costs. Equation (19) provides this equivalence by computing $d$ such that the discounted sum of per-step costs matches the episode-level bound $D$. Thus, the per-step constraint is a normalized restatement of the original CMDP constraint, rather than an additional restriction. Without this normalization, an agent could satisfy all per-step SCVaR conditions while still violating the episodic limit, leading to unstable or inconsistent optimization behavior. Our aim is therefore not to tighten the constraint, but to express the same safety requirement in a form compatible with SCVaR-based per-step updates. This formulation is consistent with WCSAC and its official implementation. We have added a concise clarification in the corresponding part of the main text (line 318).

---

> ### Author Response · Authors · 2025-11-28
> **(3/3)**
>
> **W7. Figure choices**
>
> **Reviewer’s concern.**
> Line plots with confidence bands may better show training cost trends than boxplots.
>
> **Re_W7.** We thank the reviewer for the suggestion to use line plots with confidence bands to visualize the evolution of training costs. We chose boxplots because training cost exhibits high variance and heavy-tailed fluctuations, and smoothing or averaging single trajectories can mask the underlying distributional structure of violations. Boxplots more clearly capture the full dispersion and tail behavior across different training phases, which are essential for assessing safety-critical performance. Our visualization style also follows the presentation used in the CAL paper. In addition, the middle row of Figure 5 already provides a line plot with confidence bands, where the progression of cost over training can be clearly observed.
>
> ---
>
> **W8. Interpretation of $\alpha$**
>
> **Reviewer’s concern.**
> $\alpha = 0.1$ yields lower rewards and raises doubts about the statement that it achieves a “precise balance,” suggesting that the choice of $\alpha$ may be task dependent.
>
> **Re_W8.** When we state that the model “achieves precise balance,” we do not mean that $\alpha = 0.1$ maximizes reward. In the Ant environment in particular, this refers to keeping SCVaR closely aligned with the safety threshold $d$ without exceeding it, thereby enforcing the safety constraint with high fidelity rather than aiming solely for higher reward. As the reviewer correctly notes, the reward–risk trade-off is environment dependent and shaped by the form of the underlying safety-cost distribution, and therefore the most suitable $\alpha$ naturally varies across tasks. Our results reflect this behavior: $\alpha = 0.5$ often produces a more favorable reward–cost compromise, while $\alpha = 0.1$ prioritizes strict constraint satisfaction. We have also reorganized the discussion in Section 4.1 to more clearly convey this task-dependent relationship between $\alpha$ and the reward–risk balance.
>
>
>
> **W9. Missing $\alpha$ in ablation**
>
> **Re_W9.**
> We thank the reviewer for pointing out this omission. All ablation studies were conducted with a fixed $\alpha = 0.5$. This mid-range value provides a balanced level of risk sensitivity and serves as a consistent setting across all ablation experiments. We have updated the text to explicitly state this for clarity（Section 4.2）.
>
> ---
>
> We thank the reviewer again for the thoughtful and constructive feedback.
> We hope these detailed clarifications address all concerns and improve the clarity and rigor of the submission.

---

### Author Response · Authors · 2025-11-30
**The Summary Comment to the AC**

**Dear Area Chair,**

We sincerely thank you for the detailed evaluation. To facilitate your quick understanding, we summarize below the strengths recognized by the reviewers, the concerns raised, and how our revision fully resolves them.

**Contributions**

Our work introduces a new distributional perspective to Safe RL by modeling long-term safety costs with GMMs and proposing SCVaR, a coherent and conservative risk measure that captures worst-case tail risks across mixture components. We develop an incremental EM-based refinement operator for stable online GMM updates and integrate this safety critic into SAC, forming GMM-SSAC, which achieves improved safety and competitive returns on standard benchmarks.

**Strengths**

Across reviews, several positive aspects of our work were consistently highlighted:

1. **Novelty and conceptual contribution.**
Reviewers highlighted the new safety critic and risk metric (SCVaR), and the use of GMMs to model multimodal costs, offering a fresh perspective beyond unimodal CVaR-based critics.

2. **Clear problem motivation and relevance.**
Reviewers agreed that our problem formulation is coherent and well motivated, and that the method integrates cleanly into off-policy SAC.

3. **Improved readability and presentation quality.**
Reviewers commented that the manuscript is overall clearly written. Following reviewer suggestions, we enhanced figure readability, refined notation, simplified the main diagram, and polished the writing throughout.

4. **Strong empirical results.**
Reviewers appreciated the breadth of tasks and stability of our method. Results consistently show better safety and reduced constraint violations.

**Weaknesses and Our Improvements**

Most concerns focused on clarity, completeness of baselines, or missing references, rather than flaws in the core method.

1. **Missing baselines and references (Reviewer fALM, score 2).**
The reviewer expressed concern that baseline selection may be biased.

**Re:**
* The missing baselines mentioned by the reviewer are all based on PPO-style on-policy algorithms, which are not aligned with our off-policy SAC-based framework.
* Nevertheless, to fully address the reviewer’s concern and demonstrate the generality of SCVaR, we added extensive PPO-based baselines, including several major on-policy risk-aware algorithms.
* We also incorporated all missing references and clarified our methodological positioning in the related-work section.

2. **Clarity issues in notation, derivations, and figure organization.**
Reviewers requested improved clarity for the dual update rule, the definition of SCVaR, and the main pipeline diagram.

**Re:**
* We corrected and clarified mathematical expressions throughout the paper.
* We significantly simplified Figure 3 while preserving technical rigor.
* We improved explanations of SCVaR, incremental refinement, and the EM-Bellman operator.

3. **Regarding generality and theoretical grounding.**
Reviewers asked whether SCVaR generalizes beyond GMMs and whether it provides concrete advantages over CVaR.

**Re:**
* We strengthened the theoretical discussion on SCVaR’s conservativeness and showed that component-wise tail selection satisfies coherence properties.
* New experiments validate alignment with empirical distributions and demonstrate advantages over unimodal Gaussian CVaR (WCSAC) and mean-based critics (SAC-Lag).

4. **Complexity, efficiency, and parameter analyses.**
* We added both theoretical justifications and empirical efficiency studies.
* We conducted extensive ablations, including whether adjusting CVaR α can match SCVaR, sensitivity of SCVaR to α, and feasibility of dynamic α schedules.

All raised issues were related to clarity and completeness and do not affect the methodology. Despite the tight rebuttal window, we significantly expanded the manuscript and added new experiments to fully address every reviewer concern.

**Regarding the Low Score and Concerns about Bias**

We respectfully clarify the concerns from the reviewer fALM.
* The concerns centered on missing PPO-based baselines and certain references.
* Our introduction follows a coherent, problem-driven narrative: it motivates Safe RL in high-risk domains, identifies the inadequacy of unimodal Gaussian critics, motivates GMM-based modeling and SCVaR, and leads naturally to an incremental EM-based safety critic integrated into SAC.
* This structure reflects focused methodological development rather than omission of unrelated work. It is not biased.
* SAC and PPO differ fundamentally, and we selected SAC-family baselines for fairness, not bias.
* Nevertheless, we reproduced the reviewer-suggested PPO baselines and added corresponding experiments.

This additional effort required substantial engineering and computational resources, contributing to a later rebuttal submission, but our intention was to provide a rigorous and responsible response.

Thank you again for your valuable time and careful consideration.

---

### Meta-Review · Area_Chair_hG4R · 2025-12-21

**Summary:**

**Summary**: The paper proposes GMM-SSAC, a safe RL framework that models the safety-cost distribution using Gaussian Mixture Models (GMMs) and introduces Supremum Conditional Value-at-Risk (SCVaR), the maximum CVaR across $K$ components, as a conservative risk metric. The safety critic is trained through a Bellman-consistent incremental EM update, while the actor minimizes an SAC-style objective penalized by SCVaR. Theoretical sections show that SCVaR upper-bounds the mixture CVaR and is a coherent risk measure; experiments on Safety-Gymnasium and velocity-constrained MuJoCo tasks show improved constraint satisfaction with comparable reward.

**Reviewers' Concerns**:  Reviewers have appreciated the new formulations and insights. In particular, the limitation of the standard CVaR and how the SCVaR proposes to address these limitations. However, they also raised several concerns. The AC has gone over these in detail and identified several key points. (i) Reviewers raised some concerns regarding the motivations behind the SCVaR. In particular, the main concern is that the experimental sections did not consider any examples that demand multimodal distributions. (ii) The paper also transforms the episode-wise constraint to per constraint, which seems to be overly conservative according to the reviewers. (iii) The lack of good baselines is also a major concern for the reviewers. The authors tried to address those concerns.


**AC's take**: The AC has gone over the paper and mostly agrees with the reviewers' concerns. Unfortunately, there were not many discussions among the authors and the reviewers; hence, there is no way to know whether the authors' rebuttals have fully addressed the concerns. The AC has gone over the rebuttals, and it seems that not all the concerns have been resolved. In particular, a few things are not clear. First, the main motivation seems to be that if one takes the CVaR of the total cost, it would certainly be less than the maximum of the CVaR values across different costs. Hence, the formulation in the paper makes perfect sense. However, some of the responses alluded to the fact that they consider the CVaR of the mixture policy. Hence, the main motivation behind the problem formulation seems to be confusing. Further, the experimental section did not consider multiple constraints to score the justification of the above consideration. The main problem requires satisfying the constraint in a cumulative sense. However, the paper tries to pose it as a step-wise constraint, which is naturally a conservative approach. There are papers that consider the per-step constraints [A1,A2], which are more conservative compared to the CMDP, as their goal is more towards the satisfaction of the safety at every step rather than over an episode. Hence, it might happen that this paper's bounds are conservative. Overall, the connection with the original paper is missing. Finally, the policy for CVaR needs to be history-dependent; however, because of the per-step constraint satisfaction, it seems that this policy is not, which might be related to the dynamic risk measure rather than the static risk measure. I would suggest that the authors should look into this to make the paper complete.


Overall, I believe that the paper is not ready for publication yet. I would encourage the authors to address the limitations.

[A1]. Amani, S., Thrampoulidis, C. and Yang, L., 2021, July. Safe reinforcement learning with linear function approximation. In International Conference on Machine Learning (pp. 243-253). PMLR.

[A2]. Roknilamouki, A., Ghosh, A., Shi, M., Nourzad, F., Ekici, E. and Shroff, N., Provably Efficient RL for Linear MDPs under Instantaneous Safety Constraints in Non-Convex Feature Spaces. In Forty-second International Conference on Machine Learning.

**Reviewer Concerns:**

Reviewers have raised several concerns, as I have pointed out. I believe that some of the concerns remain unresolved. They are: (i) Reviewers raised some concerns regarding the motivations behind the SCVaR. In particular, the main concern is that the experimental sections did not consider any examples that demand multimodal distributions. (ii) The paper also transforms the episode-wise constraint to per constraint, which seems to be overly conservative according to the reviewers. (iii) The lack of good baselines is also a major concern for the reviewers. The authors tried to address those concerns. However, there was no discussion; hence, the AC has to estimate whether the rebuttals resolved the concerns. According to the AC, the concerns have not been resolved.

**Reviewer Scores:**

Reviewers' scores are mixed. There was not much discussion, and I don't believe that they would be going to change.

---

### Decision · Program_Chairs · 2026-01-26

Reject